# Open the Black Box: Step-based Policy Updates for Temporally-Correlated Episodic Reinforcement Learning

**Ge Li**[1,*]    **Hongyi Zhou**[1]    **Dominik Roth**[1]    **Serge Thilges**[1]    **Fabian Otto**[2,3]

**Rudolf Lioutikov**[1]    **Gerhard Neumann**[1]

[1]Karlsruhe Institute of Technology, Germany    [2]University of Tübingen, Germany

[3]Bosch Center for Artificial Intelligence, Germany

## Abstract

Current advancements in reinforcement learning (RL) have predominantly focused on learning step-based policies that generate actions for each perceived state. While these methods efficiently leverage step information from environmental interaction, they often ignore the temporal correlation between actions, resulting in inefficient exploration and unsmooth trajectories that are challenging to implement on real hardware. Episodic RL (ERL) seeks to overcome these challenges by exploring in parameters space that capture the correlation of actions. However, these approaches typically compromise data efficiency, as they treat trajectories as opaque *black boxes*. In this work, we introduce a novel ERL algorithm, Temporally-Correlated Episodic RL (TCE), which effectively utilizes step information in episodic policy updates, opening the 'black box' in existing ERL methods while retaining the smooth and consistent exploration in parameter space. TCE synergistically combines the advantages of step-based and episodic RL, achieving comparable performance to recent ERL methods while maintaining data efficiency akin to state-of-the-art (SoTA) step-based RL. Our code is available at https://github.com/BruceGeLi/TCE_RL.

## 1 Introduction

By considering how policies interact with the environment, reinforcement learning (RL) methodologies can be classified into two distinct categories: step-based RL (SRL) and episodic RL (ERL). SRL predicts actions for each perceived state, while ERL selects an entire behavioral sequence at the start of an episode. Most predominant deep RL methods, such as PPO (Schulman et al., 2017) and SAC (Haarnoja et al., 2018a), fall into the category of SRL. In these methods, the step information — comprising state, action, reward, subsequent state, and done signal received by the RL agent at each discrete time step — is pivotal for policy updates. This granular data aids in estimating the policy gradient (Williams, 1992; Sutton et al., 1999), approximating state or state-action value functions (Haarnoja et al., 2018a), and assessing advantages (Schulman et al., 2015b). Although SRL methods have achieved great success in various domains, they often face significant exploration challenges. Exploration in SRL, often based on a stochastic policy like a factorized Gaussian, typically lacks temporal and cross-DoF (degrees of freedom) correlations. This deficiency leads to inconsistent and inefficient exploration across state and action spaces (Raffin et al., 2022; Schumacher et al., 2023), as shown in Fig. 1a. Furthermore, the high variance in trajectories generated through such exploration can cause suboptimal convergence and training instability, a phenomenon highlighted by considerable performance differences across various random seeds (Agarwal et al., 2021).

**Episodic RL**, in contrast to SRL, represents a distinct branch of RL that emphasizes the maximization of returns over entire episodes (Whitley et al., 1993; Igel, 2003; Peters & Schaal, 2008), rather than focusing on the internal evolution of the environment interaction. This approach shifts the solution search from per-step actions to a parameterized trajectory space, employing techniques like

---

*Corresponding author. Email to <geli.bruce.ai@gmail.com, ge.li@kit.edu>

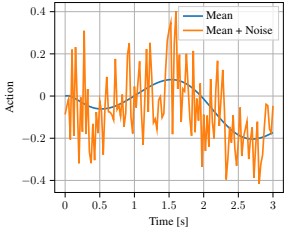 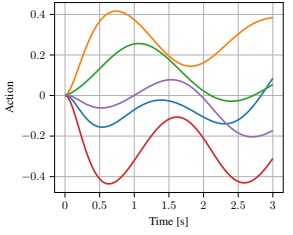 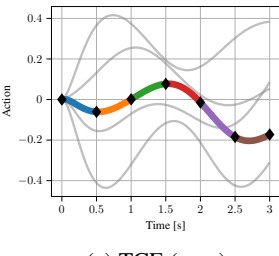

(a) Step-based RL        (b) Traj.-based, Episodic RL        (c) TCE (ours)

Figure 1: Illustration of exploration strategies: (a) SRL samples actions by adding noise to the predicted mean, resulting in inconsistent exploration and jerky actions. However, their leverage of step-based information leads to efficient policy updates. (b) ERL samples complete trajectories in a parameter space and generate consistent control signals. Yet, they often treat trajectories as single data points and overlook the step-based information during the interaction, causing inefficient policy update. (c) TCE combines the benefits of both, using per-step information for policy update while sampling complete trajectories with broader exploration and high smoothness.

Movement Primitives (MPs) (Schaal, 2006; Paraschos et al., 2013). Such exploration strategy allows for broader exploration horizons and ensures consistent trajectory smoothness across task episodes, as illustrated in Fig. 1b. Additionally, it is theoretically capable of capturing temporal correlations and interdependencies among DoF. ERL typically treats entire trajectories as single data points, often overlooking the internal changes in the environment and state transitions. This approach leads to training predominantly using black-box optimization methods (Salimans et al., 2017; Tangkaratt et al., 2017; Celik et al., 2022; Otto et al., 2022). The term *black box* in our title reflects this reliance on black-box optimization, which tends to overlook detailed step-based information acquired during environmental interactions. However, this often results in a lack of attention to the individual contributions of each segment of the trajectory to the overall task success. Consequently, while ERL excels in expansive exploration and maintaining trajectory smoothness, it typically requires a larger volume of samples for effective policy training. In contrast, step-based RL methods have demonstrated notable advancements in learning efficiency by utilizing this detailed step-based information.

**Open the Black Box.** In this paper, our goal is to integrate step-based information into the policy update process of ERL. Our proposed method, Temporally-Correlated Episodic RL (TCE), moves beyond the traditional approach of treating an entire trajectory as a single data point. Instead, we transform trajectory-wide elements, such as reproducing likelihood and advantage, into their segment-wise counterparts. This enables us to leverage the step-based information to recognize and accentuate the unique contributions of each trajectory segment to overall task success. Through this innovative approach, we have opened the black box of ERL, making it more effective while retaining its strength. As a further step, we explore the benefits of fully-correlated trajectory exploration in deep ERL. We demonstrate that leveraging full covariance matrices for trajectory distributions significantly improves policy quality in existing black-box ERL methods like Otto et al. (2022).

**Our contributions** are summarized as: (a) We propose TCE, a novel RL framework that integrates step-based information into the policy updates of ERL, while preserving the broad exploration scope and trajectory smoothness characteristic of ERL. (b) We provide an in-depth analysis of exploration strategies that effectively capture both temporal and degrees of freedom (DoF) correlations, demonstrating their beneficial impact on policy quality and trajectory smoothness. (c) We conduct a comprehensive evaluation of our approach on multiple simulated robotic manipulation tasks, comparing its performance against other baseline methods.

## 2 PRELIMINARIES

### 2.1 EPISODIC REINFORCEMENT LEARNING

**Markov Decision Process (MDP).** We consider a MDP problem of a policy search defined by a tuple $(\mathcal{S}, \mathcal{A}, \mathcal{T}, \mathcal{R}, \mathcal{P}_0, \gamma)$. We assume the state space $\mathcal{S}$ and action space $\mathcal{A}$ are continuous and the transition probabilities $\mathcal{T} : \mathcal{S} \times \mathcal{S} \times \mathcal{A} \to [0, 1]$ describe the state transition probability to $\boldsymbol{s}_{t+1}$, given the current state $\boldsymbol{s}_t \in \mathcal{S}$ and action $\boldsymbol{a}_t \in \mathcal{A}$. The initial state distribution is denoted as $\mathcal{P}_0 : \mathcal{S} \to$

$[0, 1]$. The reward $r_t(s_t, a_t)$ returned by the environment is given by a function $\mathcal{R} : \mathcal{S} \times \mathcal{A} \to \mathbb{R}$ and $\gamma \in [0, 1]$ describes the discount factor. The goal of RL in general is to find a policy $\pi$ that maximizes the expected accumulated reward, namely return, as $R = \mathbb{E}_{\mathcal{T}, \mathcal{P}_0, \pi}[\sum_{t=0}^{\infty} \gamma^t r_t]$.

**Episodic RL** (Whitley et al., 1993) focuses on maximizing the return $R = \sum_{t=0}^{T}[\gamma^t r_t]$ over a task episode of length $T$, irrespective of the state transitions within the episode. This approach typically employs a parameterized trajectory generator, like MPs (Schaal, 2006), to predict a trajectory parameter vector $w$. This vector is then used to generate a complete reference trajectory $y(w) = [y_t]_{t=0:T}$. The resulting trajectory is executed using a trajectory tracking controller to accomplish the task. In this context, $y_t \in \mathbb{R}^D$ denotes the trajectory value at time $t$ for a system with $D$ DoF, differentiating it from the per-step action $a$ used in SRL. It is important to note that, although ERL predicts an entire action trajectory, it still maintains the *Markov Property*, where the state transition probability depends only on the given current state and action (Sutton & Barto, 2018). In this respect, the action selection process in ERL is fundamentally similar to techniques like action repeat (Braylan et al., 2015) and temporally correlated action selection (Raffin et al., 2022; Eberhard et al., 2022). In contrast to SRL, ERL predicts the trajectory parameters as $\pi(w|s)$, which shifts the solution search from the per-step action space $\mathcal{A}$ to the parameter space $\mathcal{W}$. Therefore, a trajectory parameterized by a vector $w$ is typically treated as a single data point in $\mathcal{W}$. Consequently, ERL commonly employs black-box optimization methods for problem-solving (Salimans et al., 2017; Otto et al., 2022). The general learning objective of ERL is formally expressed as

$$J = \int \pi_{\theta}(w|s)[R(s, w) - V^{\pi}(s)]dw = \mathbb{E}_{w \sim \pi_{\theta}(w|s)}[A(s, w)], \tag{1}$$

where $\pi_{\theta}$ represents the policy, parameterized by $\theta$, e.g. using NNs. The initial state $s \in \mathcal{S}$ characterizes the starting configuration of the environment and the task goal, serving as the input to the policy. The $\pi_{\theta}(w|s)$ indicates the likelihood of selecting the trajectory parameter $w$. The term $R(s, w) = \sum_{t=0}^{T}[\gamma^t r_t]$ represents the return obtained from executing the trajectory, while $V^{\pi}(s) = \mathbb{E}_{w \sim \pi_{\theta}(w|s)}[R(s, w)]$ denotes the expected return across all possible trajectories under policy $\pi_{\theta}$. Their subtraction is defined as the advantage function $A(s, w)$, which quantifies the benefit of selecting a specific trajectory. By using parameterized trajectory generators like MPs, ERL benefits from consistent exploration, smooth trajectories, and robustness against local optima, as noted by Otto et al. (2022). However, its policy update strategy incurs a trade-off in terms of learning efficiency, as valuable step-based information is overlooked during policy updates. Furthermore, existing method like Bahl et al. (2020); Otto et al. (2022) commonly use factorized Gaussian policies, which inherently limits their capacity to capture all relevant movement correlations.

## 2.2 USING MOVEMENT PRIMITIVES FOR TRAJECTORY REPRESENTATION

The Movement Primitives (MP), as a parameterized trajectory generator, play an important role in ERL and robot learning. This section highlights key MP methodologies and their mathematical foundations, deferring a more detailed discussion to Appendix B. Schaal (2006) introduced the Dynamic Movement Primitives (DMPs) method, incorporating a force signal into a dynamical system to produce smooth trajectories from given initial robot states. Following this, Paraschos et al. (2013) developed Probabilistic Movement Primitives (ProMPs), which leverages a linear basis function representation to map parameter vectors to trajectories and their corresponding distributions. The probability of observing a trajectory $[y_t]_{t=0:T}$ given a specific weight vector distribution $p(w) \sim \mathcal{N}(w|\mu_w, \Sigma_w)$ is represented as a linear basis function model:

$$[y_t]_{t=0:T} = \Phi_{0:T}^{\mathsf{T}} w + \epsilon_y, \tag{2}$$
$$p([y_t]_{t=0:T}; \mu_y, \Sigma_y) = \mathcal{N}(\Phi_{0:T}^{\mathsf{T}} \mu_w, \Phi_{0:T}^{\mathsf{T}} \Sigma_w \Phi_{0:T} + \sigma_y^2 I). \tag{3}$$

Here, $\epsilon_y$ is zero-mean white noise with variance $\sigma_y^2$. The matrix $\Phi_{0:T}$ houses the basis functions for each time step $t$. Additionally, $p([y_t]_{t=0:T}; \mu_y, \Sigma_y)$ defines the trajectory distribution coupling the DoF and time steps, mapped from $p(w)$. For a $D$-DoF system with $N$ parameters per DoF and $T$ time steps, the dimensions of the variables in Eq. (2) and 3 are as follows: $w, \mu_w : D \cdot N$; $\Sigma_w : D \cdot N \times D \cdot N$; $\Phi_{0:T} : D \cdot N \times D \cdot T$; $y, \mu_y : D \cdot T$; $\Sigma_y : D \cdot T \times D \cdot T$.

Recently, Li et al. (2023) introduced Probabilistic Dynamic Movement Primitives (ProDMPs), a hybrid approach that blends the pros of both methods. Similar to ProMP, ProDMPs defines a trajectory

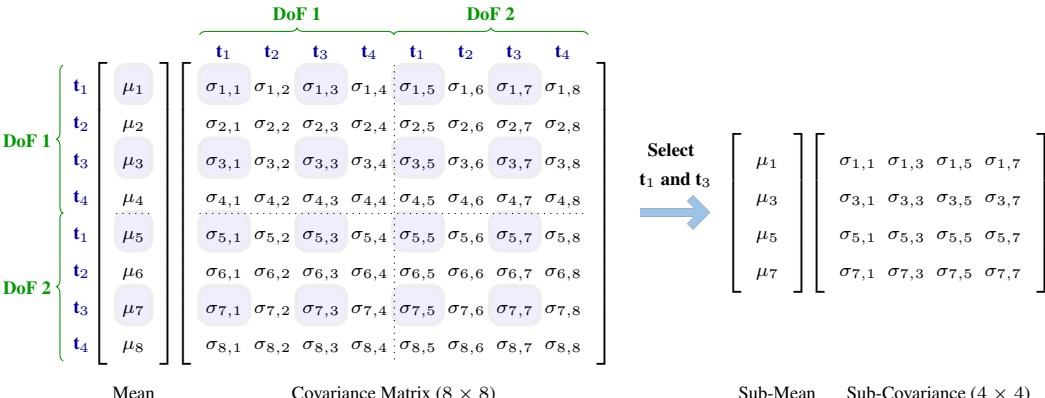

Figure 2: Reduce the trajectory distribution dimensions using two time steps (Li et al., 2023), shown in an element-wise format. Here, the trajectory has two DoF and four time steps, with $D \cdot T = 8$. **Left**: The 8-dim mean vector and the $8 \times 8$-dim covariance matrix of the original trajectory distribution, capture correlations across both DoF and time steps. **Right**: Randomly selecting two time points, e. g. $t_1$ and $t_3$, yields a reduced distribution while still capturing the movement correlations.

as $y(t) = \mathbf{\Phi}(t)^{\mathsf{T}} \boldsymbol{w} + c_1 y_1(t) + c_2 y_2(t)$. The added terms $c_1 y_1(t) + c_2 y_2(t)$ are included to ensure accurate trajectory initialization. This formulation combines the distributional modeling benefits of ProMP with the precision in trajectory initiation offered by DMP.

## 2.3 REPRESENTATION OF TRAJECTORY DISTRIBUTION AND LIKELIHOOD

Computing the trajectory distribution and reconstruction likelihood is crucial for policy updates in ERL. Previous methods like Bahl et al. (2020); Otto et al. (2022) represented the trajectory distribution using the parameter distribution $p(\boldsymbol{w})$ and the likelihood of a sampled trajectory $\boldsymbol{y}^*$ with its parameter vector as $p(\boldsymbol{w}^*|\boldsymbol{\mu_w}, \sigma_{\boldsymbol{w}}^2)$. However, this approach treats an entire trajectory as a singular data point and fails to efficiently utilize step-based information. In contrast, research in imitation learning, including works by Paraschos et al. (2013); Gomez-Gonzalez et al. (2016), maps parameter distributions to trajectory space and allows the exploitation of trajectory-specific information. Yet, the likelihood computation in this space is computationally intensive, primarily due to the need to invert a high-dimensional covariance matrix, a process with an $O((D \cdot T)^3)$ time complexity. Recent studies, like those by (Seker et al., 2019; Akbulut et al., 2021; Przystupa et al., 2023), advocates for directly modeling the trajectory distribution using neural networks. These methods typically employ a factorized Gaussian distribution $\mathcal{N}(y|\mu_y, \sigma_y^2)$, instead of a full Gaussian distribution $\mathcal{N}(\boldsymbol{y}|\boldsymbol{\mu_y}, \boldsymbol{\Sigma_y})$ that accounts for both the DoF and time steps. This choice mitigates the computational burden of likelihood calculations, but comes at the cost of neglecting key temporal correlations and interactions between different DoF. To address these challenges, Li et al. (2023) introduced a novel approach for estimating the trajectory likelihood with a set of paired time points $(t_k, t_k')$, $k = 1, ..., K$, as

$$\log p([\boldsymbol{y}_t]_{t=0:T}) \approx \frac{1}{K} \sum_{k=1}^{K} \log \mathcal{N}(\boldsymbol{y}_{(t_k, t_k')}|\boldsymbol{\mu}_{(t_k, t_k')}, \boldsymbol{\Sigma}_{(t_k, t_k')}), \tag{4}$$

As shown in Fig. 2, this method scales down the dimensions of a trajectory distribution from $D \cdot T$ to a more manageable $D \cdot 2$. Through the use of batched, randomly selected time pairs during training, the method is proved to efficiently capture correlations while reducing computational cost.

## 2.4 USING TRUST REGIONS FOR STABLE POLICY UPDATE

In ERL, the parameter space $\mathcal{W}$ typically exhibits higher dimensionality compared to the action space $\mathcal{A}$. This complexity presents unique challenges in maintaining stable policy updates. Trust regions methods (Schulman et al., 2015a; 2017) has long been recognized as an effective technique for ensuring the stability and convergence of policy gradient methods. While popular methods such as PPO approximate trust regions using surrogate cost functions, they lack the capacity for exact enforcement. To tackle this issue, Otto et al. (2021) introduced trust region projection layer (TRPL), a mathematically rigorous and scalable technique that precisely enforces trust regions in deep RL

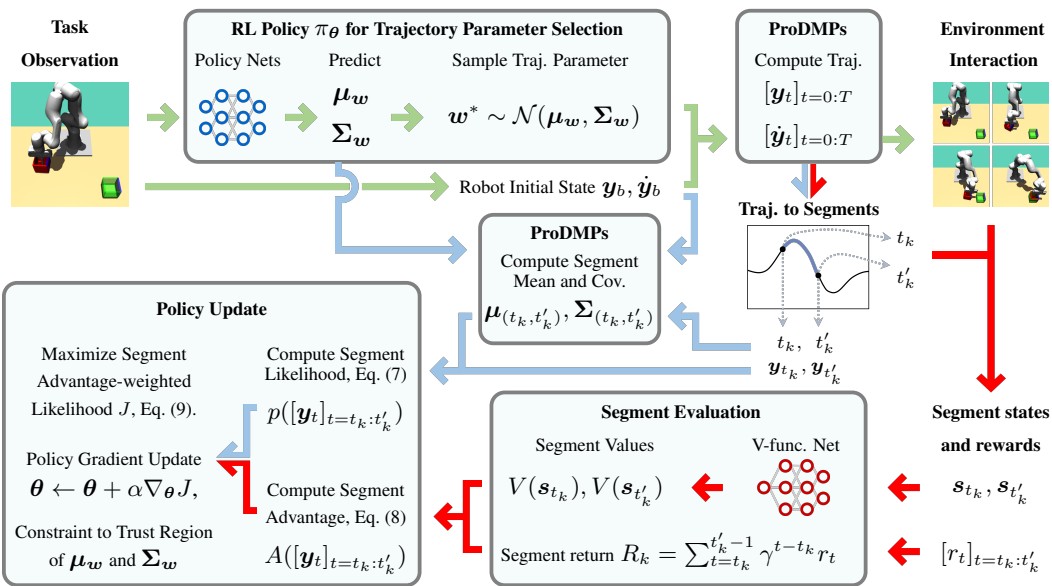

Figure 3: The TCE framework. The entire learning framework can be divided into three main parts. The first part, shown in **green** arrows, involves trajectory sampling, generation, and execution, detailing how the robot is controlled to complete a given task. The second part, indicated in **blue** arrows, focuses on estimating the likelihood of selecting a particular segment of the sampled trajectory. The third part, marked by **red** arrows, deals with segment evaluation and advantage computation, assessing how much each segment contributes to the successful task completion.

algorithms. By incorporating differentiable convex optimization layers (Agrawal et al., 2019), this method not only allows for trust region enforcement for each input state, but also demonstrates significant effectiveness and stability in high-dim parameter space, as validated in method like BBRL Otto et al. (2022). The TRPL takes standard outputs of a Gaussian policy—namely, the mean vector $\boldsymbol{\mu}$ and covariance matrix $\boldsymbol{\Sigma}$ —and applies a state-specific projection operation to maintain trust regions. The adjusted Gaussian policy, parameterized by $\tilde{\boldsymbol{\mu}}$ and $\tilde{\boldsymbol{\Sigma}}$, forms the basis for subsequent computations. Let $d_{\text{mean}}$ and $d_{\text{cov}}$ be the dissimilarity measures, e. g. KL-divergence, for mean and covariance, bounded by $\epsilon_\mu$ and $\epsilon_\Sigma$ respectively. The optimization for each state $\boldsymbol{s}$ is formulated as:

$$\arg\min_{\tilde{\boldsymbol{\mu}}_s} d_{\text{mean}}\left(\tilde{\boldsymbol{\mu}}_s, \boldsymbol{\mu}(\boldsymbol{s})\right), \quad \text{s.t.} \quad d_{\text{mean}}\left(\tilde{\boldsymbol{\mu}}_s, \boldsymbol{\mu}_{\text{old}}(\boldsymbol{s})\right) \leq \epsilon_\mu, \text{ and}$$
$$\arg\min_{\tilde{\boldsymbol{\Sigma}}_s} d_{\text{cov}}\left(\tilde{\boldsymbol{\Sigma}}_s, \boldsymbol{\Sigma}(\boldsymbol{s})\right), \quad \text{s.t.} \quad d_{\text{cov}}\left(\tilde{\boldsymbol{\Sigma}}_s, \boldsymbol{\Sigma}_{\text{old}}(\boldsymbol{s})\right) \leq \epsilon_\Sigma. \tag{5}$$

# 3 USE STEP-BASED INFORMATION FOR ERL POLICY UPDATES

We introduce an innovative framework of ERL that builds on traditional ERL foundations, aiming to facilitate an efficient policy update mechanism while preserving the intrinsic benefits of ERL. The key innovation lies in redefining the role of trajectories in the policy update process. In contrast to previous methods which consider an entire trajectory as a single data point, our approach breaks down the trajectory into individual segments. Each segment is evaluated and weighted based on its distinct contribution to the task success. This method allows for a more effective use of step-based information in ERL. The comprehensive structure of this framework is depicted in Figure 3.

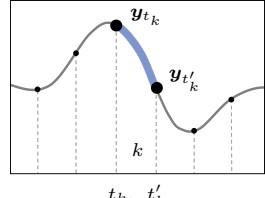

Figure 4: Divide a trajectory into $K$ segments

**Trajectory Prediction and Generation.** As highlighted by green arrows in Fig. 3, we adopt a structure similar to previous ERL works, such as the one described by Otto et al. (2022). However, this part distinguish itself by using the most recent ProDMPs for trajectory generation and distribution

modeling, due to the improved support for trajectory initialization. Additionally, we enhance the previous framework by using a full covariance matrix policy $\pi(\boldsymbol{w}|\boldsymbol{s}) = \mathcal{N}(\boldsymbol{w}|\boldsymbol{\mu_w}, \boldsymbol{\Sigma_w})$ as opposed to a factorized Gaussian policy, to capture a broader range of movement correlations.

**Trajectory Likelihood Representation.** In RL, the likelihood of previously sampled actions, along with their associated returns, is often used to adjust the chance of these actions being selected in future policies. In previous ERL methods, this process typically involves the probability of choosing an entire trajectory. However, our framework adopts a different strategy, as shown in blue arrows in Fig. 3. Using the techniques in Sections 2.2 and 2.3, our approach begins by selecting K paired time steps. We then transform the parameter likelihood into a trajectory likelihood, which is calculated using these K pairwise likelihoods. This approach, depicted in Figure 4, effectively divides the whole trajectory into K distinct segments, with each segment defined by a pair of time steps. In essence, this method allows us to break down the overall trajectory likelihood into individual segment likelihoods, offering a more detailed view of the trajectory's contribution to task success.

$$\text{Trajectory to Segments:} \qquad p([\boldsymbol{y}_t]_{t=0:T}|\boldsymbol{s}) \triangleq \{p([\boldsymbol{y}_t]_{t=t_k:t'_k}|\boldsymbol{s})\}_{k=1\ldots K}, \qquad (6)$$

$$\text{Local Representation:} \qquad p([\boldsymbol{y}_t]_{t=t_k:t'_k}|\boldsymbol{s}) \triangleq p([\boldsymbol{y}_{t_k}, \boldsymbol{y}_{t'_k}]|\boldsymbol{\mu}_{(t_k,t'_k)}(\boldsymbol{s}), \boldsymbol{\Sigma}_{(t_k,t'_k)}(\boldsymbol{s})). \qquad (7)$$

**Definition of Segment Advantages.** Similar to standard SRL methods, we leverage the advantage function to evaluate the benefits of executing individual segments within a sampled trajectory. When being at state $\boldsymbol{s}_{t_k}$ and following the trajectory segment $[\boldsymbol{y}_t]_{t=t_k:t'_k}$, the segment-wise advantage function $A(\boldsymbol{s}_{t_k}, [\boldsymbol{y}_t]_{t=t_k:t'_k})$ quantifies the difference between the actual return obtained by executing this sampled trajectory segment and the expected return from a randomly chosen segment, as

$$A(\boldsymbol{s}_{t_k}, [\boldsymbol{y}_t]_{t=t_k:t'_k}) = \sum_{t=t_k}^{t=t'_k-1} \gamma^{t-t_k} r_t + \gamma^{t'_k-t_k} V^{\pi_{\text{old}}}(\boldsymbol{s}_{t'_k}) - V^{\pi_{\text{old}}}(\boldsymbol{s}_{t_k}), \qquad (8)$$

where $V^{\pi_{\text{old}}}(\boldsymbol{s}_{t_k})$ denotes the value function of the current policy. In our method, the estimation of $V^{\pi_{\text{old}}}(\boldsymbol{s}_{t_k})$ is consistent with the approaches commonly used in SRL and is independent of the design choice of segment selections. We use NNs to estimate $V^{\pi}(\boldsymbol{s}) \approx V^{\pi}_{\phi}(\boldsymbol{s})$ which is fitted on targets of true return or obtained by general advantage estimation (GAE) (Schulman et al., 2015b).

**Learning Objective.** By replacing the trajectory likelihood and advantage with their segment-based counterparts in the original ERL learning objective as stated in Eq. (1), we propose the learning objective of our method as follows

$$J(\boldsymbol{\theta}) = \mathbb{E}_{\pi_{\text{old}}} \left[ \frac{1}{K} \sum_{k=1}^{K} \frac{p_{\pi_{\text{new}}}([\boldsymbol{y}_t]_{t=t_k:t'_k}|\boldsymbol{s})}{p_{\pi_{\text{old}}}([\boldsymbol{y}_t]_{t=t_k:t'_k}|\boldsymbol{s})} A^{\pi_{\text{old}}}(\boldsymbol{s}_{t_k}, [\boldsymbol{y}_t]_{t=t_k:t'_k}) \right]. \qquad (9)$$

Here, $\boldsymbol{s}$ denotes the initial state of the episode, used for selecting the parameter vector $\boldsymbol{w}$, and $\boldsymbol{s}_{t_k}$ is the state of the system at time $t_k$. The learning objective takes the *Importance Sampling* to update policies using data from previous policies (Schulman et al., 2015a; 2017; Otto et al., 2021). Our method retains the advantages of exploration in parameter space and generating smooth trajectories. This enables us to enhance the likelihood of segments with high advantage and reduce the likelihood of less rewarding ones during policy updates. To ensure a stable update for the full covariance Gaussian policy $\pi_{\boldsymbol{\theta}}(\boldsymbol{w}|\boldsymbol{s}) = \mathcal{N}(\boldsymbol{\mu_w}, \boldsymbol{\Sigma_w})$, we deploy a differentiable Trust Region Projection step (Otto et al., 2021) after each policy update iteration as previously discussed in Section 2.4.

## 4 RELATED WORKS

**Improve Exploration and Smoothness in Step-based RL.** SRL methods, such as PPO and SAC, interact with the environment by performing a sampled action at each time-step. This strategy often results in a control signal with high-frequency noise, making it unsuitable for direct use in robotic systems (Raffin et al., 2022). A prevalent solution is to reduce the sampling frequency, a technique commonly known as *frame skip* (Braylan et al., 2015). Here, the agent only samples actions every k-th time step and replicates this action for the skipped steps. Similar approaches decide whether to repeat the last action or to sample a new action in every time step (Biedenkapp et al., 2021; Yu et al., 2021). This concept is also echoed in works such as general State Dependent Exploration

(gSDE) (Raffin et al., 2022; Rückstieß et al., 2008; Chiappa et al., 2023), where the applied noise is sampled in a state-dependent fashion; leading to smooth changes of the disturbance between consecutive steps. However, while these methods improved the smoothness in small segments, they struggled to model long-horizon correlations. Another area of concern is the utilization of white noise during sampling, which fails to consider the temporal correlations between time steps and results in a random walk with suboptimal exploration. To mitigate this, previous research, such as Lillicrap et al. (2015) and Eberhard et al. (2022), have integrated colored noise into the RL policy, aiming to foster exploration that is correlated across time steps. While these methods have shown advantages over white noise approaches, they neither improve the trajectory's smoothness, nor adequately capture the cross-DoF correlations.

**Episodic RL.** The early ERL approaches used black-box optimization to evolve parameterized policies, e.g., small NN (Whitley et al., 1993; Igel, 2003; Gomez et al., 2008). However, these early works were limited to tasks with low-dimensional action space, for instance, the Cart Pole. Although recent works (Salimans et al., 2017; Mania et al., 2018) have shown that, with more computing, these methods can achieve comparable asymptotic performance with step-based algorithms in some locomotion tasks, none of these methods can deal with tasks with context variations (e.g., changing goals). Another research line in ERL works with more compact policy representation. Peters & Schaal (2008); Kober & Peters (2008) first combined ERL with MPs, reducing the dimension of searching space from NN parameter space to MPs parameter space with the extra benefits of smooth trajectories generation. Abdolmaleki et al. (2015) proposed a model-based method to improve the sample efficiency. Notably, although those methods can already handle some complex manipulation tasks such as robot baseball (Peters & Schaal, 2008), none of them can deal with contextual tasks. To deal with that problem, (Abdolmaleki et al., 2017) further extends that utilizes linear policy conditioned on the context. Another recent work from this research line (Celik et al., 2022) proposed using a Mixture of Experts (MoE) to learn versatile skills under the ERL framework.

**BBRL.** As the first method that utilizes non-linear adaptation to contextual ERL, Deep Black Box Reinforcement Learning (BBRL) (Otto et al., 2022) is the most related work to our method. BBRL applies trust-region-constrained policy optimization to learn a weight adaptation policy for MPs. Despite demonstrating great success in learning tasks with sparse and non-Markovian rewards, it requires significantly more samples to converge compared to SoTA SRL methods. This could be attributed to its black-box nature, where the trajectory from the entire episode is treated as a single data point, and the trajectory return is calculated by summing up all step rewards within the episode.

## 5 EXPERIMENTS

We evaluate the effectiveness of our model through experiments on a variety of simulated robot manipulation tasks. The performance of TCE is compared against well-known deep RL algorithms as well as methods specifically designed for correlated actions and consistent trajectories. The evaluation is designed to answer the following questions: (a) Can our model effectively train the policy across diverse tasks, incorporating various robot types, control paradigms (task and joint space), and reward configurations? (b) Does the incorporation of movement correlations lead to higher task success rates? (c) Are there limitations or trade-offs when adopting our proposed learning strategy?

For the comparative evaluation, we select the following methods: PPO, SAC, TRPL, gSDE, PINK (Eberhard et al., 2022) and BBRL. Descriptions, hyper-parameters, and references to the used code bases of these methods can be found in Appendix C.1.We use step-based methods (PPO, SAC, TRPL, gSDE, and PINK) to predict the lower-level actions for each task. On the other hand, for episodic methods like BBRL and TCE, we predict position and velocity trajectories and then employ a PD controller to compute the lower-level control commands. Across all experiments, we measure task success in terms of the number of environment interactions required. Each algorithm is evaluated using 20 distinct random seeds. Results are quantified using the Interquartile Mean (IQM) and are accompanied by a 95% stratified bootstrap confidence interval (Agarwal et al., 2021).

### 5.1 LARGE SCALE ROBOT MANIPULATION BENCHMARK USING METAWORLD

We begin our evaluation using the Metaworld benchmark (Yu et al., 2020), a comprehensive testbed that includes 50 manipulation tasks of varying complexity. Control is executed in a 3-DOF task

space along with the finger closeness, and a dense reward signal is employed. In contrast to the original evaluation protocol, we introduce a more stringent framework. Specifically, each episode is initialized with a randomly generated context, rather than a fixed one. Additionally, we tighten the success criteria to only consider a task successfully completed if the objective is maintained until the final time step. This adjustment mitigates scenarios where transient successes are followed by erratic agent behavior. While we train separate policies for each task, the hyperparameters remain constant across all 50 tasks. For each method, we report the overall success rate as measured by the IQM across the 50 sub-tasks in Fig. 5a. The performance profiles are presented in Fig. 5b.

In both metrics, our method significantly outperforms the baselines in achieving task success. BBRL exhibits the second-best performance in terms of overall consistency across tasks but lags in training speed compared to step-based methods. We attribute this difference to the use of per-step dense rewards, which enables faster policy updates in step-based approaches. TCE leverages the advantages of both paradigms, surpassing other algorithms after approximately $10^7$ environment interactions. Notably, the off-policy

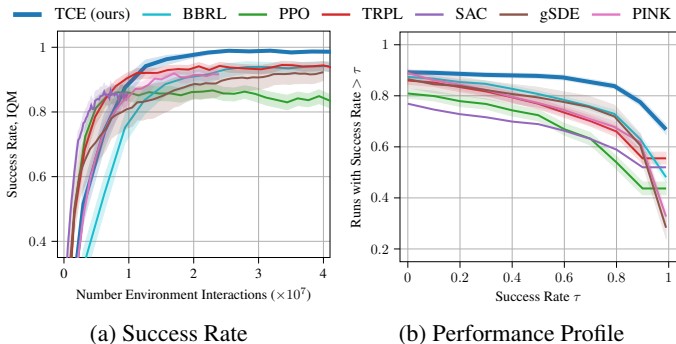

(a) Success Rate   (b) Performance Profile

Figure 5: Metaworld Evaluation. (a) Overall Success Rate across all 50 tasks, reported using Interquartile Mean (IQM) (Agarwal et al., 2021). (b) Performance profile, illustrating the fraction of runs that exceed the threshold specified on the x-axis.

methods SAC and PINK were trained with fewer samples than used for on-policy methods due to their limitations in parallel environment utilization. PINK achieved superior final performance but at the expense of sample efficiency compared to SAC. In Appendices C.2 and C.3, we provide the results for 50 tasks and a performance profile analysis of TCE.

## 5.2 JOINT SPACE CONTROL WITH MULTI TASK OBJECTIVES

Next, we investigate the advantages of modeling complete movement correlations and the utility of intermediate feedback for policy optimization. To this end, we enhance the BBRL algorithm by expanding its factorized Gaussian policy to accommodate full covariance (BBRL Cov.), thereby capturing movement correlations. Both the original and augmented versions of BBRL are included in the subsequent task evaluations. We evaluate the methods on a customized Hopper Jump task, sourced from OpenAI Gym (Brockman et al., 2016). This 3-DoF task primarily focuses on maximizing jump height while also accounting for precise landing at a designated location. Control is executed in joint space. We report the max jump height as the main metric of success in Fig. 6a. Our method exhibits quick learning and excels in maximizing jump height. Both BBRL versions exhibit similar performance, while BBRL Cov. demonstrates marginal improvements over the original. However, they both fall behind TCE in training speed, highlighting the efficiency gains we achieve through intermediate state-based policy updates. Step-based methods like PPO and TRPL tend to converge to sub-optimal policies. The only exception is gSDE. As an augmented step-based method, it enables smoother and more consistent exploration but exhibits significant sensitivity to model initialization (random seeds), evident from the wide confidence intervals.

## 5.3 CONTACT-RICH MANIPULATION WITH DENSE AND SPARSE REWARD SETTINGS

We further turn to a 7-DoF robot box-pushing task adapted from (Otto et al., 2022). The task requires the robot's end-effector, equipped with a rod, to maneuver a box to a specified target position and orientation. The difficulty lies in the need for continuous, correlated movements to both position and orient the box accurately. To amplify the complexity, the initial pose of the box is randomized. We test two reward settings: dense and sparse. The dense setting offers intermediate rewards inversely proportional to the current distance between the box and its target pose, while the sparse setting only allocates rewards at the episode's end based on the final task state. Performance metrics for both settings are shown in Fig. 6b and 6c. In either case, TCE and gSDE exhibit superior performance but

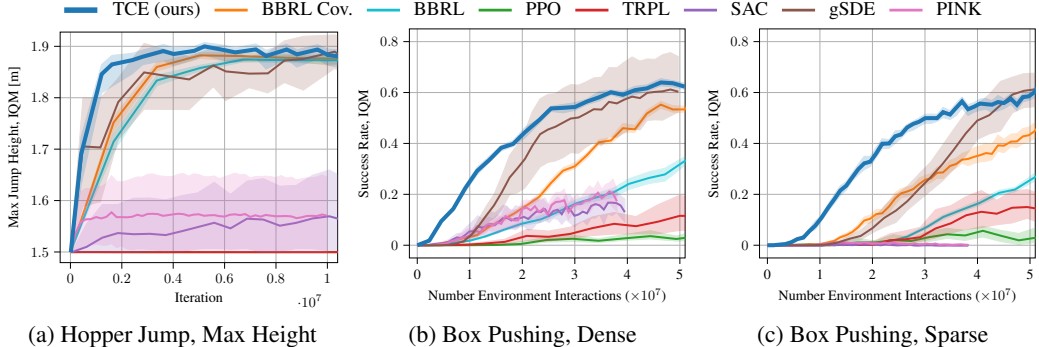

Figure 6: Task Evaluation of (a) Hopper Jump Max Height. (b) Box Pushing success rate in dense reward, and (c) Box Pushing success rate in sparse reward setting.

with TCE demonstrating greater consistency across different random seeds. The augmented BBRL version outperforms its original counterpart, emphasizing the need for fully correlated movements in tasks that demand consistent object manipulation. The other step-based methods struggle to learn the task effectively, even when dense rewards are provided. This further highlights the advantages of modeling the movement trajectory as a unified action, as opposed to a step-by-step approach.

## 5.4 HITTING TASK WITH HIGH SPARSITY REWARD SETTING

In our last experiment, we assess the limitations of our method using a 7-DoF robot table tennis task, originally from (Celik et al., 2022). The robot aims to return a randomly incoming ball to a desired target on the opponent's court. To enhance the task's realism, we randomize the robot's initial state instead of using a fixed starting pose. This task is distinct due to its one-shot nature: the robot has only one chance to hit the ball and loses control over the ball's trajectory thereafter. The need for diverse hitting strategies like forehand and backhand adds complexity and increases the number of samples required for training. Performance metrics are presented in Fig. 7. The BBRL Cov. emerges as the leader, achieving a 20% higher success rate than other methods. It is followed by TCE and the original BBRL, with TCE displaying intermediate learning speeds between the two BBRL versions. Step-based methods, led by TRPL at a mere 15% task success, struggle notably in this setting. We attribute the underperformance of TCE and step-based methods to the task's reward sparsity, which complicates the value function learning of SRL and TCE. Despite these challenges, TCE maintains its edge over other baselines, further attesting to its robustness, even under stringent conditions.

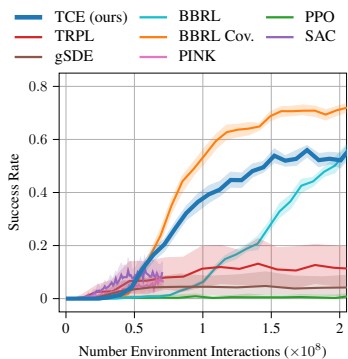

Figure 7: Table Tennis with High reward sparsity.

## 6 CONCLUSION

We introduced TCE that synergizes the exploration advantages of ERL with the sample efficiency of SRL. Empirical evaluation showcases that TCE matches the sample efficiency of SRL and consistently delivers competitive asymptotic performance across various tasks. Furthermore, we demonstrated both the sample efficiency and policy performance of episodic policies can be further improved by incorporating proper correlation modeling. Despite the promise, several opening questions remain for future work. Firstly, TCE yields moderate results for tasks characterized by particularly sparse reward settings, as observed in scenarios like table tennis. Secondly, ERL approaches often need a low-level tracking controller, which might not be feasible for certain task types, such as locomotion. Additionally, the current open-loop control setup lacks the adaptability needed for complex control problems in dynamic environments where immediate feedback and swift adaptation are crucial. These issues will be at the forefront of our future work.

ACKNOWLEDGMENTS

We thank our colleagues Onur Celik, Maximilian Xiling Li, Vaisakh Kumar Shaj, and Balázs Gyenes at KIT for the valuable discussion, technical support and proofreading. We thank the anonymous reviewers for their insightful feedback which greatly improved the quality of this paper.

The research presented in this paper was funded by the Deutsche Forschungsgemeinschaft (DFG, German Research Foundation) – 448648559, and was supported in part by the Helmholtz Association of German Research Centers. Gerhard Neumann was supported in part by Carl Zeiss Foundation through the Project JuBot (Jung Bleiben mit Robotern). The authors acknowledge support by the state of Baden-Württemberg through bwHPC, and the HoreKa supercomputer.

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

# List of Content in Appendix

## A ALGORITHM BOX

---

**Algorithm 1** Temporally-Correlated Episodic RL (TCE)

---

1: Initialize policy parameters $\theta$ and value function parameters $\phi$
2: **for** iteration = 1, 2, ... **do**
3:  Get the initial state $s_0$
4:  Predict the mean $\mu_w$, covariance $\Sigma_w$, and sample $w*$
5:  Generate the trajectory $y*$ using Eq. (2) and execute it in the environment
6:  Collect step-based information through the execution
7:  Update $\phi$, use true return or GAE style return (Schulman et al., 2015b)
8:
9:  Select $K$ time-pairs, e.g. choose every 10 steps along the trajectory
10:  Compute the segment-wise likelihood $\{p_k^{\text{old}}\}_{k=1:K}$ using Eq. (6) and 7 under $\pi^{\text{old}}$
11:  **for** update epoch = 1, 2, ... **do**
12:   Make prediction of the mean $\mu_w^{\text{new}}$, covariance $\Sigma_w^{\text{new}}$ under the latest policy $\pi^{\text{new}}$
13:   Enforce Trust Region by projecting $\mu_w^{\text{new}}$ and $\Sigma_w^{\text{new}}$ through TRPL using Eq. (5)
14:   Get projected policy $\tilde{\pi}^{\text{new}}$, represented by $\tilde{\mu}_w^{\text{new}}$ and $\tilde{\Sigma}_w^{\text{new}}$
15:   Compute the segment-wise likelihood $\{p_k^{\text{new}}\}_{k=1:K}$ using Eq. (6) and 7 under $\tilde{\pi}^{\text{new}}$
16:   Update $\theta$ by taking a gradient step w.r.t. $\hat{J}(\theta)$ in Eq. (9)
17:  **end for**
18: **end for**

---

# B   MATHEMATICAL FORMULATIONS OF MOVEMENT PRIMITIVES.

In this section, we provide a concise overview of the mathematical formulations of movement primitives utilized in this paper. We begin with the fundamentals of DMPs and ProMPs, followed by a detailed presentation of ProDMPs. This includes a focus on trajectory computation and the mapping between parameter distributions and trajectory distributions. For clarity, we begin with a single DoF system and then present the full trajectory distribution using a multi-DoF systems.

## B.1   DYNAMIC MOVEMENT PRIMITIVES (DMPS)

Schaal (2006); Ijspeert et al. (2013) describe a single movement as a trajectory $[y_t]_{t=0:T}$, which is governed by a second-order linear dynamical system with a non-linear forcing function $f$. The mathematical representation is given by

$$\tau^2 \ddot{y} = \alpha(\beta(g - y) - \tau\dot{y}) + f(x), \quad f(x) = x \frac{\sum \varphi_i(x) w_i}{\sum \varphi_i(x)} = x \boldsymbol{\varphi}_x^{\mathsf{T}} \boldsymbol{w}, \tag{10}$$

where $y = y(t)$, $\dot{y} = \mathrm{d}y/\mathrm{d}t$, $\ddot{y} = \mathrm{d}^2y/\mathrm{d}t^2$ denote the position, velocity, and acceleration of the system at a specific time $t$, respectively. Constants $\alpha$ and $\beta$ are spring-damper parameters, $g$ signifies a goal attractor, and $\tau$ is a time constant that modulates the speed of trajectory execution. To ensure convergence towards the goal, DMPs employ a forcing function governed by an exponentially decaying phase variable $x(t) = \exp(-\alpha_x/\tau; t)$. Here, $\varphi_i(x)$ represents the basis functions for the forcing term. The trajectory's shape as it approaches the goal is determined by the weight parameters $w_i \in \boldsymbol{w}$, for $i = 1, ..., N$. The trajectory $[y_t]_{t=0:T}$ is typically computed by numerically integrating the dynamical system from the start to the end point (Pahič et al., 2020; Bahl et al., 2020). However, this numerical process is computationally intensive, and complicates a directly translation between a parameter distribution $p(w)$ to its corresponding trajectory distribution $p(y)$ (Amor et al., 2014; Meier & Schaal, 2016). Previous method, such as GMM/GMR (Calinon et al., 2012; Calinon, 2016; Yang et al., 2018) used Gaussian Mixture Models to cover the trajectories' domain. However, this neither captures temporal correlation nor provide a generative model for the trajectories.

## B.2   PROBABILISTIC MOVEMENT PRIMITIVES (PROMPS)

Paraschos et al. (2013) introduced a framework for modeling MPs using trajectory distributions, capturing both temporal and inter-dimensional correlations. Unlike DMPs that use a forcing term, ProMPs directly model the intended trajectory. The probability of observing a 1-DoF trajectory $[y_t]_{t=0:T}$ given a specific weight vector distribution $p(\boldsymbol{w}) \sim \mathcal{N}(\boldsymbol{w}|\boldsymbol{\mu_w}, \boldsymbol{\Sigma_w})$ is represented as a linear basis function model:

$$\text{Linear basis function:} \quad [y_t]_{t=0:T} = \boldsymbol{\Phi}_{0:T}^{\mathsf{T}} \boldsymbol{w} + \epsilon_y, \tag{11}$$

$$\text{Mapping distribution:} \quad p([y_t]_{t=0:T}; \boldsymbol{\mu_y}, \boldsymbol{\Sigma_y}) = \mathcal{N}(\boldsymbol{\Phi}_{0:T}^{\mathsf{T}} \boldsymbol{\mu_w}, \ \boldsymbol{\Phi}_{0:T}^{\mathsf{T}} \boldsymbol{\Sigma_w} \boldsymbol{\Phi}_{0:T} + \sigma_y^2 \boldsymbol{I}). \tag{12}$$

Here, $\epsilon_y$ is zero-mean white noise with variance $\sigma_y^2$. The matrix $\boldsymbol{\Phi}_{0:T}$ houses the basis functions for each time step $t$. Similar to DMPs, these basis functions can be defined in terms of a phase variable instead of time. ProMPs allows for flexible manipulation of MP trajectories through probabilistic operators applied to $p(\boldsymbol{w})$, such as conditioning, combination, and blending (Maeda et al., 2014; Gomez-Gonzalez et al., 2016; Shyam et al., 2019; Rozo & Dave, 2022; Zhou et al., 2019). However, ProMPs lack an intrinsic dynamic system, which means they cannot guarantee a smooth transition from the robot's initial state or between different generated trajectories.

## B.3   PROBABILISTIC DYNAMIC MOVEMENT PRIMITIVES (PRODMPS)

**Solving the ODE underlying DMPs**   Li et al. (2023) noted that the governing equation of DMPs, as specified in Eq. (10), admits an analytical solution. This is because it is a second-order linear non-homogeneous ODE with constant coefficients. The original ODE and its homogeneous counterpart can be expressed in standard form as follows:

$$\text{Non-homo. ODE:} \quad \ddot{y} + \frac{\alpha}{\tau}\dot{y} + \frac{\alpha\beta}{\tau^2}y = \frac{f(x)}{\tau^2} + \frac{\alpha\beta}{\tau^2}g \equiv F(x, g), \tag{13}$$

$$\text{Homo. ODE:} \quad \ddot{y} + \frac{\alpha}{\tau}\dot{y} + \frac{\alpha\beta}{\tau^2}y = 0. \tag{14}$$

The solution to this ODE is essentially the position trajectory, and its time derivative yields the velocity trajectory. These are formulated as:

$$y = \begin{bmatrix} y_2\boldsymbol{p_2} - y_1\boldsymbol{p_1} & y_2 q_2 - y_1 q_1 \end{bmatrix} \begin{bmatrix} \boldsymbol{w} \\ g \end{bmatrix} + c_1 y_1 + c_2 y_2 \tag{15}$$

$$\dot{y} = \begin{bmatrix} \dot{y}_2\boldsymbol{p_2} - \dot{y}_1\boldsymbol{p_1} & \dot{y}_2 q_2 - \dot{y}_1 q_1 \end{bmatrix} \begin{bmatrix} \boldsymbol{w} \\ g \end{bmatrix} + c_1 \dot{y}_1 + c_2 \dot{y}_2. \tag{16}$$

Here, the learnable parameters $\boldsymbol{w}$ and $g$ which control the shape of the trajectory, are separable from the remaining terms. Time-dependent functions $y_1, y_2, \boldsymbol{p}_1, \boldsymbol{p}_2, q_1, q_2$ in the remaining terms offer the basic support to generate the trajectory. The functions $y_1, y_2$ are the complementary solutions to the homogeneous ODE presented in Eq. (14), and $\dot{y}_1, \dot{y}_2$ their time derivatives respectively. These time-dependent functions take the form as:

$$y_1(t) = \exp\left(-\frac{\alpha}{2\tau}t\right), \qquad\qquad y_2(t) = t\exp\left(-\frac{\alpha}{2\tau}t\right), \tag{17}$$

$$\boldsymbol{p}_1(t) = \frac{1}{\tau^2}\int_0^t t'\exp\left(\frac{\alpha}{2\tau}t'\right)x(t')\boldsymbol{\varphi}_x^\mathsf{T}\mathrm{d}t', \qquad \boldsymbol{p}_2(t) = \frac{1}{\tau^2}\int_0^t \exp\left(\frac{\alpha}{2\tau}t'\right)x(t')\boldsymbol{\varphi}_x^\mathsf{T}\mathrm{d}t', \tag{18}$$

$$q_1(t) = \left(\frac{\alpha}{2\tau}t - 1\right)\exp\left(\frac{\alpha}{2\tau}t\right) + 1, \qquad q_2(t) = \frac{\alpha}{2\tau}\left[\exp\left(\frac{\alpha}{2\tau}t\right) - 1\right]. \tag{19}$$

It's worth noting that the $\boldsymbol{p}_1$ and $\boldsymbol{p}_2$ cannot be analytically derived due to the complex nature of the forcing basis terms $\boldsymbol{\varphi}_x$. As a result, they need to be computed numerically. Despite this, isolating the learnable parameters, namely $\boldsymbol{w}$ and $g$, allows for the reuse of the remaining terms across all generated trajectories. These residual terms can be more specifically identified as the position and velocity basis functions, denoted as $\boldsymbol{\Phi}(t)$ and $\dot{\boldsymbol{\Phi}}(t)$, respectively. When both $\boldsymbol{w}$ and $g$ are included in a concatenated vector, represented as $\boldsymbol{w}_g$, the expressions for position and velocity trajectories can be formulated in a manner akin to that employed by ProMPs:

$$\textbf{Position:} \quad y(t) = \boldsymbol{\Phi}(t)^\mathsf{T}\boldsymbol{w}_g + c_1 y_1(t) + c_2 y_2(t), \tag{20}$$

$$\textbf{Velocity:} \quad \dot{y}(t) = \dot{\boldsymbol{\Phi}}(t)^\mathsf{T}\boldsymbol{w}_g + c_1 \dot{y}_1(t) + c_2 \dot{y}_2(t). \tag{21}$$

**In the main paper, for simplicity and notation convenience, we use $w$ instead of $w_g$ to describe the parameters and goal of ProDMPs.**

**Smooth trajectory transition** The coefficients $c_1$ and $c_2$ serve as solutions to the initial value problem delineated by the Eq.(20)(21). Li et al. propose utilizing the robot's initial state or the replanning state, characterized by the robot's position and velocity $(y_b, \dot{y}_b)$ to ensure a smooth commencement or transition from a previously generated trajectory. Denote the values of the complementary functions and their derivatives at the condition time $t_b$ as $y_{1_b}, y_{2_b}, \dot{y}_{1_b}$ and $\dot{y}_{2_b}$. Similarly, denote the values of the position and velocity basis functions at this time as $\boldsymbol{\Phi}_b$ and $\dot{\boldsymbol{\Phi}}_b$ respectively. Using these notations, $c_1$ and $c_2$ can be calculated as follows:

$$\begin{bmatrix} c_1 \\ c_2 \end{bmatrix} = \begin{bmatrix} \frac{\dot{y}_{2_b}y_b - y_{2_b}\dot{y}_b}{y_{1_b}\dot{y}_{2_b} - y_{2_b}\dot{y}_{1_b}} + \frac{y_{2_b}\dot{\boldsymbol{\Phi}}_b^\mathsf{T} - \dot{y}_{2_b}\boldsymbol{\Phi}_b^\mathsf{T}}{y_{1_b}\dot{y}_{2_b} - y_{2_b}\dot{y}_{1_b}}\boldsymbol{w}_g \\ \frac{y_{1_b}\dot{y}_b - \dot{y}_{1_b}y_b}{y_{1_b}\dot{y}_{2_b} - y_{2_b}\dot{y}_{1_b}} + \frac{\dot{y}_{1_b}\boldsymbol{\Phi}_b^\mathsf{T} - y_{1_b}\dot{\boldsymbol{\Phi}}_b^\mathsf{T}}{y_{1_b}\dot{y}_{2_b} - y_{2_b}\dot{y}_{1_b}}\boldsymbol{w}_g \end{bmatrix}. \tag{22}$$

Substituting Eq. (22) into Eq. (20) and Eq. (21), the position and velocity trajectories take the form as

$$y = \xi_1 y_b + \xi_2 \dot{y}_b + [\xi_3 \boldsymbol{\Phi}_b + \xi_4 \dot{\boldsymbol{\Phi}}_b + \boldsymbol{\Phi}]^\mathsf{T}\boldsymbol{w}_g, \tag{23}$$

$$\dot{y} = \dot{\xi}_1 y_b + \dot{\xi}_2 \dot{y}_b + [\dot{\xi}_3 \boldsymbol{\Phi}_b + \dot{\xi}_4 \dot{\boldsymbol{\Phi}}_b + \dot{\boldsymbol{\Phi}}]^\mathsf{T}\boldsymbol{w}_g \tag{24}$$

Here, $\xi_k$ for $k \in \{1, 2, 3, 4\}$ serve as intermediate terms that are derived from the complementary functions and the initial conditions. The formations of these terms are elaborated below. To find their derivatives $\dot{\xi}_k$, one can simply replace $y_1, y_2$ with their time derivatives $\dot{y}_1, \dot{y}_2$ in the equations.

$$\xi_1(t) = \frac{\dot{y}_{2_b}y_1 - \dot{y}_{1_b}y_2}{y_{1_b}\dot{y}_{2_b} - y_{2_b}\dot{y}_{1_b}}, \qquad \xi_2(t) = \frac{y_{1_b}y_2 - y_{2_b}y_1}{y_{1_b}\dot{y}_{2_b} - y_{2_b}\dot{y}_{1_b}},$$

$$\xi_3(t) = \frac{\dot{y}_{1_b}y_2 - \dot{y}_{2_b}y_1}{y_{1_b}\dot{y}_{2_b} - y_{2_b}\dot{y}_{1_b}}, \qquad \xi_4(t) = \frac{y_{2_b}y_1 - y_{1_b}y_2}{y_{1_b}\dot{y}_{2_b} - y_{2_b}\dot{y}_{1_b}}.$$

**Extend to a High DoF system** Both ProMPs and ProDMPs can be generalized to accommodate high-DoF systems. This allows for the capture of both temporal correlations and interactions among various DoF. Such generalization is implemented through modifications to matrix structures and vector concatenations, as illustrated in Paraschos et al. (2013); Li et al. (2023). To be specific, the basis functions $\mathbf{\Phi}, \dot{\mathbf{\Phi}}$, along with their values at the condition time $\mathbf{\Phi}_b, \dot{\mathbf{\Phi}}_b$, are extended to block-diagonal matrices $\mathbf{\Psi}, \dot{\mathbf{\Psi}}, \mathbf{\Psi}_b$ and $\dot{\mathbf{\Psi}}_b$ respectively. This extension is executed by tiling the existing basis function matrices $D$ times along their diagonal, where $D$ is the number of DoF. Additionally, the robot initial conditions for each DoF are concatenated into one vectors. For instance, the initial positions $y_b^1, ..., y_b^D$ are unified into a single vector $\mathbf{y}_b = [y_b^1, ..., y_b^D]^\intercal$. In this way, the position and velocity trajectories are extended as

$$\mathbf{y} = \xi_1 \mathbf{y}_b + \xi_2 \dot{\mathbf{y}}_b + [\xi_3 \mathbf{\Psi}_b + \xi_4 \dot{\mathbf{\Psi}}_b + \mathbf{\Psi}]^\intercal \mathbf{w}_g, \tag{25}$$

$$\dot{\mathbf{y}} = \dot{\xi}_1 \mathbf{y}_b + \dot{\xi}_2 \dot{\mathbf{y}}_b + [\dot{\xi}_3 \mathbf{\Psi}_b + \dot{\xi}_4 \dot{\mathbf{\Psi}}_b + \dot{\mathbf{\Psi}}]^\intercal \mathbf{w}_g. \tag{26}$$

**Parameter distribution to trajectory distribution** In a manner analogous to the description provided for ProMPs from Equation Eq. (2) to Equation Eq. (3), ProDMPs also exhibits a comparable architecture framework. This similarity is particularly evident in the structure of the learnable parameters, denoted as $\mathbf{w}_g$, which follow a linear basis function format. Consequently, it is reasonable to delineate the trajectory distribution for ProDMPs in fashion akin to that of ProMPs. Given that the parameter distribution $\mathbf{w}_g$ follows a Gaussian distribution $\mathbf{w}_g \sim \mathcal{N}(\mathbf{w}_g | \boldsymbol{\mu}_{\mathbf{w}_g}, \boldsymbol{\Sigma}_{\mathbf{w}_g})$ and adhering to the probabilistic formulation analogous to ProMPs as indicated in Eq. (3), the trajectory distribution for ProDMPs can be expressed as:

$$p([y_t]_{t=0:T}; \boldsymbol{\mu}_{\mathbf{y}}, \boldsymbol{\Sigma}_{\mathbf{y}}) = \mathcal{N}([\mathbf{y}_t]_{t=0:T} | \boldsymbol{\mu}_{\mathbf{y}}, \boldsymbol{\Sigma}_{\mathbf{y}}), \tag{27}$$

where

$$\boldsymbol{\mu}_{\mathbf{y}} = \boldsymbol{\xi}_1 \mathbf{y}_b + \boldsymbol{\xi}_2 \dot{\mathbf{y}}_b + \mathbf{H}_{0:T}^\intercal \boldsymbol{\mu}_{\mathbf{w}_g}, \qquad \boldsymbol{\Sigma}_{\mathbf{y}} = \mathbf{H}_{0:T}^\intercal \boldsymbol{\Sigma}_{\mathbf{w}_g} \mathbf{H}_{0:T} + \sigma_n^2 \mathbf{I},$$

$$\mathbf{H}_{0:T} = \boldsymbol{\xi}_3 \mathbf{\Psi}_b + \boldsymbol{\xi}_4 \dot{\mathbf{\Psi}}_b + \mathbf{\Psi}_{0:T}, \qquad \boldsymbol{\xi}_k = [\xi_k(t)]_{t=0:T}.$$

In this context, the trajectory mean, denoted as $\boldsymbol{\mu}_{\mathbf{y}}$ constitutes a vector of dimension $DT$, whereas the trajectory covariance, represented by $\boldsymbol{\Sigma}_{\mathbf{y}}$ is a $DT \times DT$ dimensional matrix. These quantities serve to integrate the trajectory values across all degrees of freedom (DoF) and temporal steps, encapsulating them within a single distribution. This multi-DoF ProDMPs representation can be seen as an enhancement of the ProMPs framework, augmented by the inclusion of initial condition terms. This ensures that the trajectories sampled under this distribution start from the specified initial state. **Additionally, the time range $t = 0 : T$ is flexible and can be replaced by any set of discrete time points. For instance, in the TCE method, a pair of time points $t_k$ and $t'_k$ can define a trajectory segment, allowing for down-sampling of the trajectory distribution to specific trajectroy segment.**

## C EXPERIMENT DETAILS

### C.1 DETAILS OF METHODS IMPLEMENTATION

**PPO**    Proximal Policy Optimization (PPO) (Schulman et al., 2017) is a prominent on-policy step-based RL algorithm that refines the policy gradient objective, ensuring policy updates remain close to the behavior policy. PPO branches into two main variants: PPO-Penalty, which incorporates a KL-divergence term into the objective for regularization, and PPO-Clip, which employs a clipped surrogate objective. In this study, we focus our comparisons on PPO-Clip due to its prevalent use in the field. Our implementation of PPO is based on the implementation of Raffin et al. (2021).

**SAC**    Soft Actor-Critic (SAC) (Haarnoja et al., 2018a;b) employs a stochastic step-based policy in an off-policy setting and utilizes double Q-networks to mitigate the overestimation of Q-values for stable updates. By integrating entropy regularization into the learning objective, SAC balances between expected returns and policy entropy, preventing the policy from premature convergence. Our implementation of SAC is based on the implementation of Raffin et al. (2021).

**TRPL**    Trust Region Projection Layers (TRPL) (Otto et al., 2021), akin to PPO, addresses the problem of stabilizing the on-policy policy gradient by constraining the learning policy staying close to the behavior policy. TRPL formulates the constrained optimization problem as a projection problem, providing a mathematically rigorous and scalable technique that precisely enforces trust regions on each state, leading to stable and efficient on-policy updates. We evaluated its performance based on the implementation of the original work.

**gSDE**    Generalized State Dependent Exploration (gSDE) (Raffin et al., 2022; Rückstieß et al., 2008; Rückstiess et al., 2010) is an exploration method designed to address issues with traditional step-based exploration techniques and aims to provide smoother and more efficient exploration in the context of robotic reinforcement learning, reducing jerky motion patterns and potential damage to robot motors while maintaining competitive performance in learning tasks.

To achieve this, gSDE replaces the traditional approach of independently sampling from a Gaussian noise at each time step with a more structured exploration strategy, that samples in a state-dependent manner. The generated samples not only depend on parameter of the Gaussian distribution $\mu$ & $\Sigma$, but also on the activations of the policy network's last hidden layer ($s$). We generate disturbances $\epsilon_t$ using the equation

$$\epsilon_t = \theta_\epsilon s, \text{ where } \theta_\epsilon \sim \mathcal{N}^d (0, \Sigma) .$$

The exploration matrix $\theta_\epsilon$ is composed of vectors of length $\text{Dim}(a)$ that were drawn from the Gaussian distribution we want gSDE to follow. The vector $s$ describes how this set of pre-computed exploration vectors are mixed. The exploration matrix is resampled at regular intervals, as guided by the 'sde sampling frequency' (ssf), occurring every n-th step if n is our ssf.

gSDE is versatile, applicable as a substitute for the Gaussian Noise source in numerous on- and off-policy algorithms. We evaluated its performance in an on-policy setting using PPO by utilizing the reference implementation for gSDE from Raffin et al. (2022). In order for training with gSDE to remain stable and reach high performance the usage of a linear schedule over the clip range had to be used for some environments.

**PINK**    We utilize SAC to evaluate the effectiveness of pink noise for efficient exploration. Eberhard et al. (2022) propose to replace the independent action noise $\epsilon_t$ of

$$a_t = \mu_t + \sigma_t \cdot \epsilon_t$$

with correlated noise from particular random processes, whose power spectral density follow a power law. In particular, the use of pink noise, with the exponent $\beta = 1$ in $S(f) = |\mathcal{F}[\epsilon](f)|^2 \propto f^{-\beta}$, should be considered (Eberhard et al., 2022).

We follow the reference implementation and sample chunks of Gaussian pink noise using the inverse Fast Fourier Transform method proposed by Timmer & Koenig (1995). These noise variables are used for SAC's exploration but the the actor and critic updates sample the independent action distribution without pink noise. Each action dimension uses an independent noise process which

causes temporal correlation within each dimension but not across dimensions. Furthermore, we fix the chunk size and maximum period to 10000 which avoids frequent jumps of chunk borders and increases relative power of low frequencies.

**BBRL-Cov/Std**    Black-Box Reinforcement Learning (BBRL) (Otto et al., 2022; 2023) is a recent developed episodic reinforcement learning method. By utilizing ProMPs (Paraschos et al., 2013) as the trajectory generator, BBRL learns a policy that explores at the trajectory level. The method can effectively handle sparse and non-Markovian rewards by perceiving an entire trajectory as a unified data point, neglecting the temporal structure within sampled trajectories. However, on the other hand, BBRL suffers from relatively low sample efficiency due to its black-box nature. Moreover, the original BBRL employs a degenerate Gaussian policy with diagonal covariance. In this study, we extend BBRL to learn Gaussian policy with full covariance to build a more competitive baseline. For clarity, we refer to the original method as BBRL-Std and the full covariance version as BBRL-Cov. We integrate BBRL with ProDMPs (Li et al., 2023), aiming to isolate the effects attributable to different MP approaches.

## C.2    METAWORLD PERFORMANCE PROFILE ANALYSIS

The distribution of success rates, reported in the performance profile in Fig. 5b, may seem to contradict the nearly perfect IQM of TCE but in reality provide insight into the consistency of TCE. Nearly two thirds of runs exceed 99% success rate and are therefore able to perfectly solve the task with this seed. The individual performances reported in Appendix C.3 show that only very few tasks, e.g., Disassemble and Hammer, have a significant fraction of unsuccessful seeds. This consistency per task is also visible in the profile, as only two percent of runs fall between zero and sixty percent success rate which is visible by the near zero slope in this range. All methods are entirely unable to solve a small set of tasks and therefore show a gap in the profile. This does not contradict the very high IQM success rate of TCE, as the IQM trims the upper and lower 25% of results. The commonly reported median effectively trims 50% and would result in even higher values. Due to the smaller and later decline in the profile compared to the other methods, the intersection between 75% of runs and the profile is located at a success rate of 90%. Therefore, only a small fraction of runs, roughly ten percent, fall within the 25% trim but only slightly decrease the value of the IQM due their high success rate. Other methods have a larger fraction of imperfect runs with lower success rate within the quartiles.

## C.3 Performance on Individual Metaworld Tasks

We report the Interquartile Mean (IQM) of success rates for each Metaworld task. The plots clearly illustrate the varying levels of difficulty across different tasks.

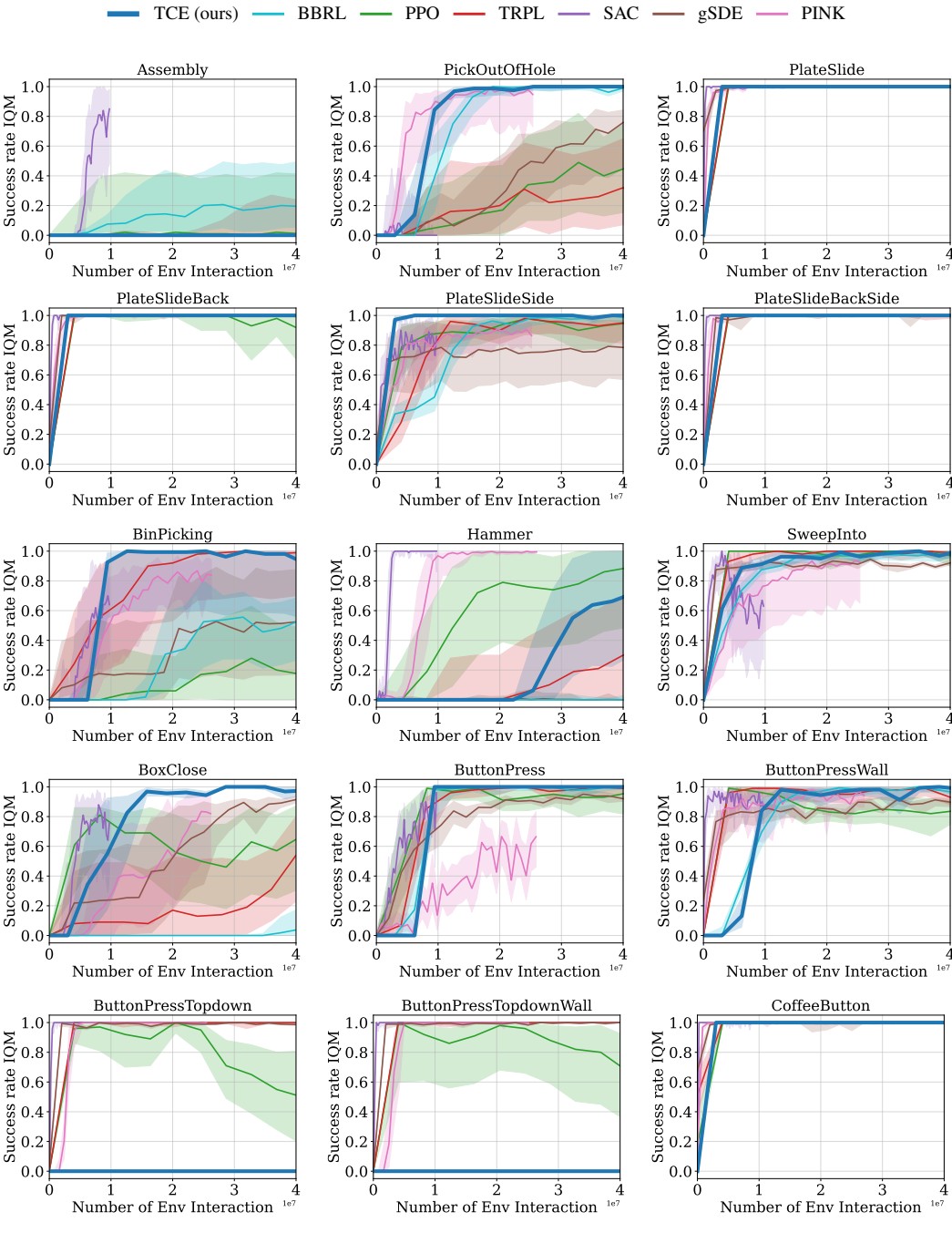

Figure 8: Success Rate IQM of each individual Metaworld tasks.

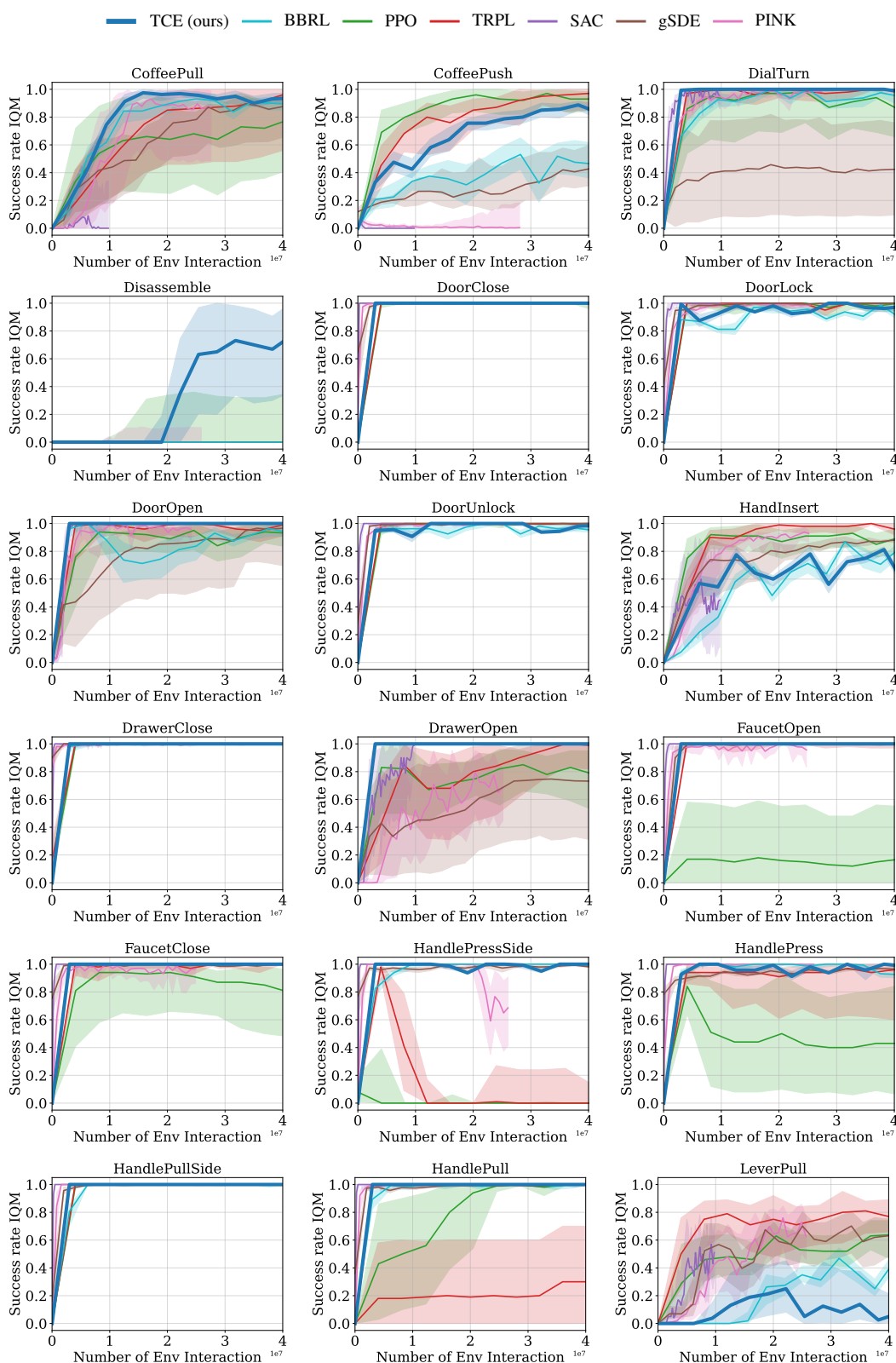

Figure 9: Success Rate IQM of each individual Metaworld tasks.

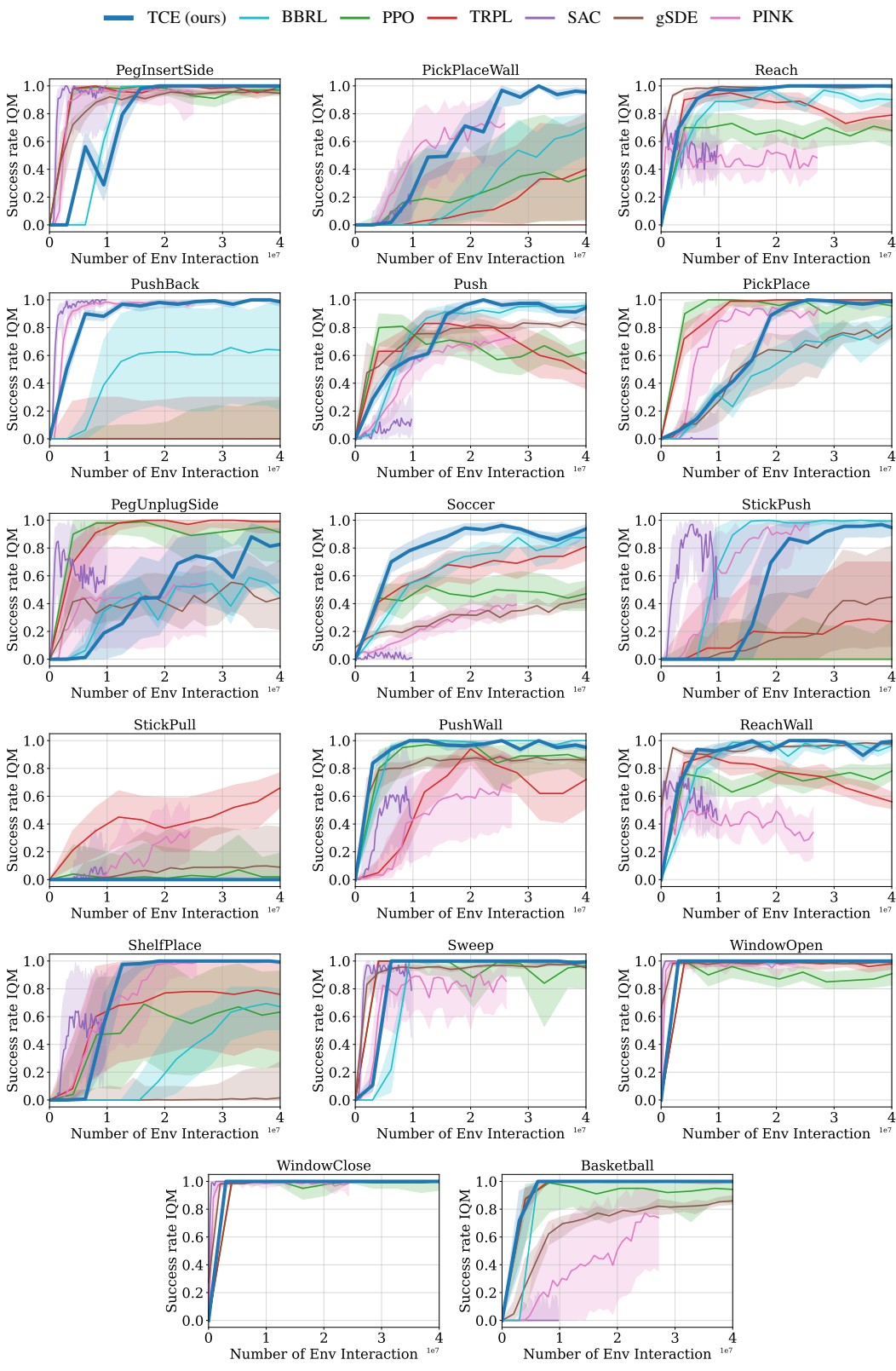

Figure 10: Success Rate IQM of each individual Metaworld tasks.

## C.4 HOPPER JUMP

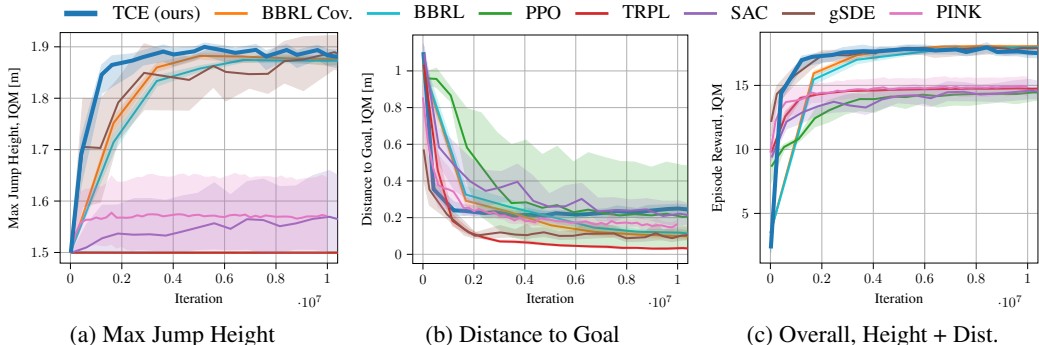

(a) Max Jump Height          (b) Distance to Goal          (c) Overall, Height + Dist.

Figure 11: Hopper Jump

As an addition to the main paper, we provide more details on the Hopper Jump task. We look at both the main goal of maximizing jump height and the secondary goal of landing on a desired position. These are shown along with the overall episode reward in Fig. 11. Our method shows quick learning and does well in achieving high jump height, consistent with what we reported earlier. While it's not as strong in landing accuracy, it still ranks high in overall performance. Both versions of BBRL have similar results. However, they train more slowly compared to TCE, highlighting the speed advantage of our method due to the use of intermediate states for policy updates. Looking at other methods, step-based ones like PPO and TRPL focus too much on landing distance and miss out on jump height, leading to less effective policies. On the other hand, gSDE performs well but is sensitive to the initial setup, as shown by the wide confidence ranges in the results. Lastly, SAC and PINK shows inconsistent results in jump height, indicating the limitations of using pink noise for exploration, especially when compared to gSDE.

## C.5 BOX PUSHING

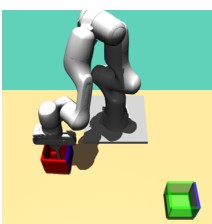

The goal of the box-pushing task is to move a box to a specified goal location and orientation using the 7-DoFs Franka Emika Panda (Otto et al., 2022). To make the environment more challenging, we extend the environment from a fixed initial box position and orientation to a randomized initial position and orientation. The range of both initial and target box pose varies from $x \in [0.3, 0.6], y \in [-0.45, 0.45], \theta_z \in [0, 2\pi]$. Success is defined as a positional distance error of less than 5 cm and a z-axis orientation error of less than 0.5 rad. We refer to the original paper for the observation and action spaces definition and the reward function.

## C.6 TABLE TENNIS

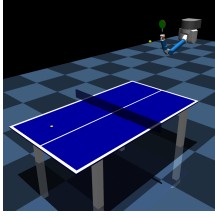

The goal of table tennis environment to use the 7-DoFs robotic arm to hit the incoming ball and return it as close as possible to the specified goal location. We adapt the table tennis environment from Celik et al. (2022); Otto et al. (2022) and extend it to a randomized initial robot joint configuration. As context space we consider the initial ball position $x \in [-1, -0.2]$, $y \in [-0.65, 0.65]$ and the goal position $x \in [-1.2, -0.2], y \in [-0.6, 0.6]$. The task is considered successful if the returned ball lands on the opponent's side of the table and within $\leq 0.2$m to the goal location. We refer to the original paper for the observation and action spaces definition and the reward function.

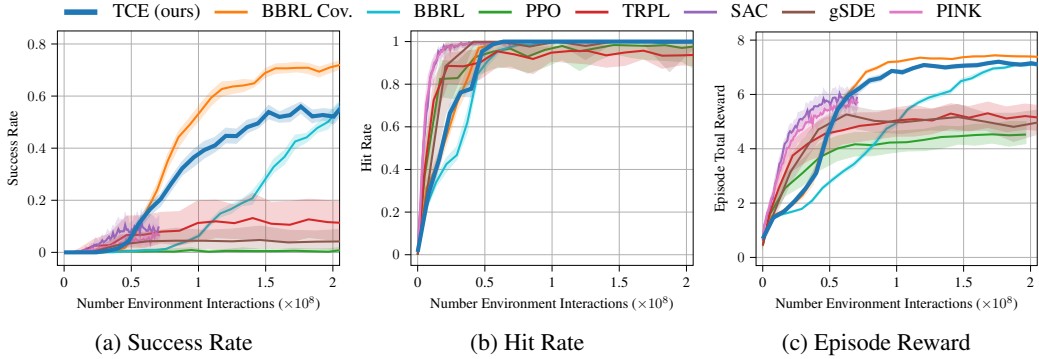

Figure 14: Robot Table Tennis Rand2Rand

# D    ADDITIONAL EVALUATION AND ABLATION STUDY

## D.1    TRAJECTORY SMOOTHNESS METRIC

We compared the trajectory smoothness of all methods in Table 1. To ensure a fair comparison, all methods were trained using the fixed start box pushing dense reward setting as originally reported in Otto et al. (2022), where each method achieved a minimum 50% success rate. Trajectories for evaluation were generated using the mean prediction of the policy. The smoothness was assessed using three metrics: *maximum jerk*, *mean squared jerk* (Wininger et al., 2009), and *dimensionless jerk* (Hogan & Sternad, 2009). The first two metrics are standard in robot trajectory generation (Berscheid & Kröger, 2021; Lange & Suppa, 2015), while the last is proposed as a more equitable measure of smoothness, eliminating the effects of motion magnitude and time scaling. TCE and BBRL Cov outperformed all other methods in smoothness, followed by the original BBRL. This performance disparity likely stems from the original BBRL's inability to model inter-DoF movement correlations. In contrast, all step-based methods exhibited lower smoothness, attributable to their inherent per-step action selection approach.

Table 1: Trajectory Smoothness, Mean (Std) of Three Metrics over 400 Trajectories.

| Metric | TCE | BBRL Cov | BBRL | PPO | TRPL | gSDE | SAC | PINK |
|---|---|---|---|---|---|---|---|---|
| **Maximum Jerk**, $\times 10^3 \mathrm{rad/s}^3$ | **3.4 (1.9)** | 3.5 (1.5) | 4.3 (1.4) | 9.1 (3.3) | 12.9 (4.8) | 6.9 (2.2) | 9.3 (4.2) | 6.5 (1.7) |
| **Mean Sq. Jerk**, $\times 10^6 \mathrm{rad}^2/\mathrm{s}^6$ | **0.2 (0.2)** | **0.2 (0.3)** | 0.6 (0.6) | 1.3 (0.9) | 5.5 (8.6) | 0.8 (0.6) | 3.9 (1.1) | 1.7 (0.7) |
| **Dimensionless Jerk**, $\times 10^6$ | **61 (73)** | 64 (56) | 128 (49) | 141 (67) | 555 (472) | 122 (83) | 506 (262) | 311 (127) |

## D.2 ACTION CORRELATIONS PREDICTED BY TRAINED POLICIES

We plot the action correlation coupling DoF and time steps in Fig. 16. All policies were trained under the box-pushing task with a dense reward setting. The action outputs for TCE, BBRL, and BBRL Cov are the positions of the robot joints, while step-based methods, such as PPO, predict actions in the torque space. TCE and BBRL Cov demonstrate the ability to predict actions correlated both temporally and across DoF, as indicated by the non-zero off-diagonal elements in their correlation matrices. In contrast, the original BBRL translates a factorized weight distribution into a block-diagonal action correlation matrix, capturing variance within individual DoF but not between them. Similarly, PINK is constrained to modeling intra-DoF correlations, which depend solely on the time difference. This limitation arises from the wide-sense stationarity of the noise, resulting in a constant value along each diagonal. gSDE, however, models temporal correlation but only over a few consecutive time steps, observable along the diagonal elements. Actions predicted by PPO, TRPL, and SAC lack both temporal and DoF correlation, resulting in correlation matrices resembling identity matrices. Interestingly, for methods that only capture intra-DoF correlations, these correlations are uniformly positive. This trend may relate to the control cost in the reward function, promoting consistent movement within each DoF over time. On the other hand, TCE and BBRL Cov are unique in their ability to capture negative correlations, both between and within DoF, enhancing their flexibility in trajectory sampling.

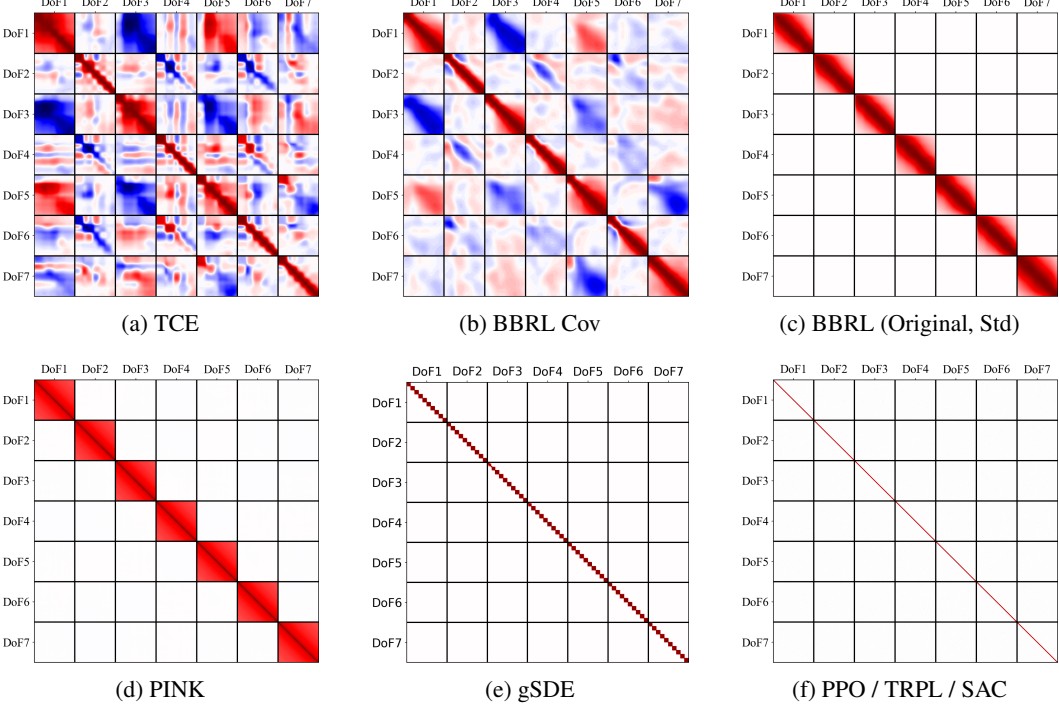

(a) TCE  (b) BBRL Cov  (c) BBRL (Original, Std)

(d) PINK  (e) gSDE  (f) PPO / TRPL / SAC

Figure 16: This figure presents predicted actions' correlation across 7 DoF and 100 time steps, visualized in a 700x700 correlation matrix. Each $100 \times 100$ square tile demonstrates the movement correlation between two DoF during these steps. Correlation values range from -1 (negative correlation, depicted in blue) to 1 (positive correlation, depicted in red), with white areas indicating no correlation. The action outputs for TCE, BBRL, and BBRL Cov are the positions of the robot joints, whereas step-based methods predict actions in the torque space. TCE and BBRL Cov exhibit to a higher capacity of movement correlations. The original BBRL and PINK only model correlations within each DoF. gSDE models correlations over a few consecutive time steps. We show only one representative matrix for PPO, TRPL, and SAC, as their results are visually identical, typically resulting in matrices resembling the identity matrix.

### D.3 Ablation: SAC + Motion Primitives-based Method

Training movement primitive-based methods using standard RL techniques, such as PPO and SAC, generally poses challenges due to the complex, higher-dimensional trajectory parameter space. In the study by Otto et al. (2022), an ablation study employing a PPO-style trust region (likelihood clipping) for training BBRL demonstrated inferior performance compared to the use of a differentiable trust region projection layer.

In Figure 17, we present an additional ablation study where SAC is used to learn the trajectory parameters of movement primitives. This study compares the performance of SAC with that of the original BBRL and BBRL Cov, leading to relatively poorer performance. The SAC model selected for reporting was the best performer among 40 different combinations of hyperparameters. The hyperparameters adjusted include the output action scaling factor (necessary because the SAC action space is bounded by $[-1, 1]$), policy/critic learning rate, batch size, and the size of the policy/critic network. The relatively shorter training curve of SAC can be attributed to its higher computational cost in policy update (Haarnoja et al., 2018b).

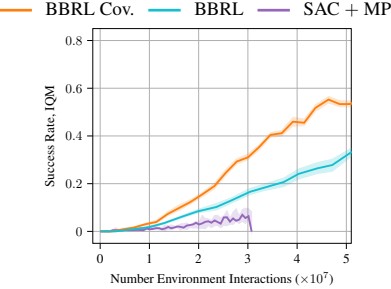

Figure 17: By employing the standard SAC method to learn trajectory parameters $w$, we compared its performance with that of the original BBRL and BBRL Cov methods under the box pushing dense reward setting. The ablated method, using SAC, showed relatively poorer performance.

### D.4 Ablation: Using PPO Style Trust Regions for TCE Method

We developed an ablated version of our method, incorporating the PPO-style trust region via likelihood clipping. We tuned the clipping ratio $\epsilon$ between 0.05 and 0.2. As illustrated in Figure 18, this version's performance falls between the original TCE and the standard PPO. The movement primitives' high-dimensional parameter space limits the effectiveness of likelihood clipping in precisely maintaining the trust region during policy updates. This limitation likely accounts for the performance gap between TCE and its PPO variant. Nonetheless, the ablated model still demonstrates a notable advantage over standard PPO, further substantiating our model's effectiveness in temporally correlated trajectory prediction.

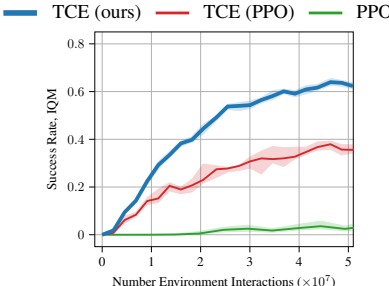

Figure 18: Training TCE with a PPO-style trust region, employing likelihood clipping with $\epsilon = 0.1$, yields suboptimal performance in the box pushing dense reward setting. Nevertheless, its superior performance compared to standard PPO underscores our method's effectiveness in episodic trajectory modeling.

## D.5  ABLATION: SELECTION OF THE AMOUNT OF SEGMENTS K

We conducted an ablation study to evaluate the effect of varying the number of segments (k) on model performance. The number of segments tested ranged from 2 to 100. Our experiments involved training in both dense and sparse box-pushing environments. The results revealed a greater sensitivity to the number of segments in the sparse reward environment compared to the dense environment. We attribute this to the challenges associated with the value function approximation under sparse reward settings. However, within an optimal range, such as 10-25 segments, this parameter is not overly sensitive compared to other hyper-parameters. Consequently, we have adopted k=25 for all experiments in this paper.

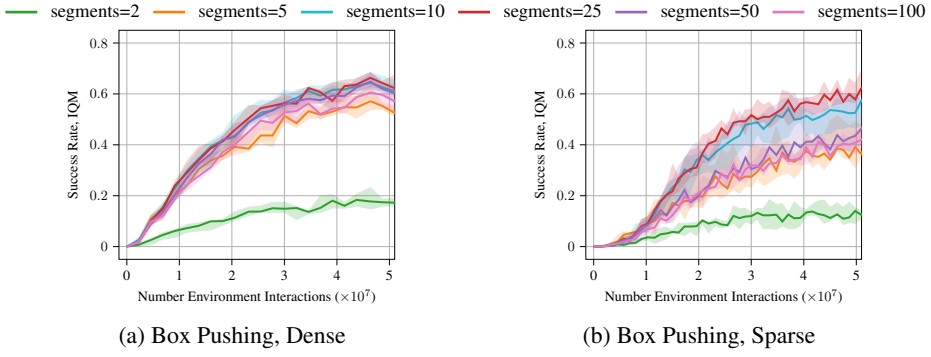

(a) Box Pushing, Dense                  (b) Box Pushing, Sparse

Figure 19: Study of the number of segment per trajectory.

## D.6  TCE COV VS. STD

We conducted an ablation study to assess the impact of employing a full covariance policy in the TCE framework. This involved comparing the standard TCE with its variant, TCE Std, which utilizes a factorized Gaussian policy $\mathcal{N}(\boldsymbol{w}|\boldsymbol{\mu_w}, \boldsymbol{\sigma_w^2})$. The comparison was conducted in scenarios involving both dense and sparse reward settings in box pushing tasks. The findings revealed that the ablated version, TCE Std, consistently underperformed compared to the full covariance version. This underperformance is attributed to the limited correlation capacity of the factorized Gaussian policy.

Furthermore, it is important to note that while the factorized Gaussian distribution results in a relatively lower computational load in the parameter space, it does not offer a marked advantage when translated into trajectory space. As illustrated in Fig. 16(c) of Section D.2, a factorized parameter distribution ultimately converts into a blocked diagonal trajectory distribution. Although this format is visually simpler compared to a full trajectory covariance matrix, both share same time complexity in terms of likelihood computation. This computational process is significantly more resource-intensive than that for a purely diagonal matrix. Therefore, we utilize the techniques in Li et al. (2023) to apply a likelihood estimation and reduce the computational cost.

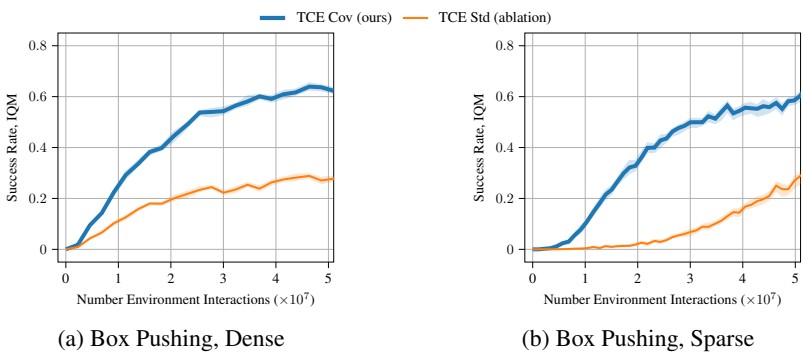

(a) Box Pushing, Dense                  (b) Box Pushing, Sparse

Figure 20: Study of the TCE Cov vs. TCE Std.

# E    HYPER PARAMETERS

We executed a large-scale grid search to fine-tune key hyperparameters for each baseline method. For other hyperparameters, we relied on the values specified in their respective original papers. Below is a list summarizing the parameters we swept through during this process.

**BBRL:**    Policy net size, critic net size, policy learning rate, critic learning rate, samples per iteration, trust region dissimilarity bounds, number of parameters per movement DoF.

**TCE:**    Same types of hyper-parameters listed in BBRL, plus the number of segments per trajectory. A learning rate decaying scheduler is applied to stabilize the training in the end.

**PPO:**    Policy network size, critic network size, policy learning rate, critic learning rate, batch size, samples per iteration.

**TRPL:**    Policy network size, critic network size, policy learning rate, critic learning rate, batch size, samples per iteration, trust region dissimilarity bounds.

**gSDE:**    Same types of hyper-parameters listed in PPO, together with the state dependent exploration sampling frequency (Raffin et al., 2022).

**SAC:**    Policy network size, critic network size, policy learning rate, critic learning rate, alpha learning rate, batch size, Update-To-Data (UTD) ratio.

**PINK:**    Same types of hyper-parameters listed in SAC.

The detailed hyper parameters used are listed in the following tables. Unless stated otherwise, the notation lin_x refers to a linear schedule. It interpolates linearly from x to 0 during training. The ERL methods (TCE, BBRL) take an entire trajectory as a sample where the SRL methods take one time step as a sample. In this way, one sample in ERL is equivlent to $T$ sample of SRL, where $T$ is the length of one task episode.

Table 2: Hyperparameters for the Meta-World experiments. Episode Length $T = 500$

| | PPO | gSDE | TRPL | SAC | PINK | TCE | BBRL |
|---|---|---|---|---|---|---|---|
| number samples | 16000 | 16000 | 16000 | 1000 | 4 | 16 | 16 |
| GAE $\lambda$ | 0.95 | 0.95 | 0.95 | n.a. | n.a. | 0.95 | n.a. |
| discount factor | 0.99 | 0.99 | 0.99 | 0.99 | 0.99 | 1 | 1 |
| | | | | | | | |
| $\epsilon_\mu$ | n.a. | n.a. | 0.005 | n.a. | n.a. | 0.005 | 0.005 |
| $\epsilon_\Sigma$ | n.a. | n.a. | 0.0005 | n.a. | n.a. | 0.0005 | 0.0005 |
| trust region loss coef. | n.a. | n.a. | 10 | n.a. | n.a. | 1 | 10 |
| | | | | | | | |
| optimizer | adam | adam | adam | adam | adam | adam | adam |
| epochs | 10 | 10 | 20 | 1000 | 1 | 50 | 100 |
| learning rate | 3e-4 | 1e-3 | 5e-5 | 3e-4 | 3e-4 | 3e-4 | 3e-4 |
| use critic | True | True | True | True | True | True | True |
| epochs critic | 10 | 10 | 10 | 1000 | 1 | 50 | 100 |
| learning rate critic | 3e-4 | 1e-3 | 3e-4 | 3e-4 | 3e-4 | 3e-4 | 3e-4 |
| number minibatches | 32 | n.a. | 64 | n.a. | n.a. | n.a. | n.a. |
| batch size | n.a. | 500 | n.a. | 256 | 512 | n.a. | n.a. |
| buffer size | n.a. | n.a. | n.a. | 1e6 | 2e6 | n.a. | n.a. |
| learning starts | 0 | 0 | 0 | 10000 | 1e5 | 0 | 0 |
| polyak_weight | n.a. | n.a. | n.a. | 5e-3 | 5e-3 | n.a. | n.a. |
| SDE sampling frequency | n.a. | 4 | n.a. | n.a. | n.a. | n.a. | n.a. |
| entropy coefficient | 0 | 0 | 0 | auto | auto | 0 | 0 |
| | | | | | | | |
| normalized observations | True | True | True | False | False | True | False |
| normalized rewards | True | True | False | False | False | False | False |
| observation clip | 10.0 | n.a. | n.a. | n.a. | n.a. | n.a. | n.a. |
| reward clip | 10.0 | 10.0 | n.a. | n.a. | n.a. | n.a. | n.a. |
| critic clip | 0.2 | lin_0.3[1] | n.a. | n.a. | n.a. | n.a. | n.a. |
| importance ratio clip | 0.2 | lin_0.3[1] | n.a. | n.a. | n.a. | n.a. | n.a. |
| | | | | | | | |
| hidden layers | [128, 128] | [128, 128] | [128, 128] | [256, 256] | [256, 256] | [128, 128] | [32, 32] |
| hidden layers critic | [128, 128] | [128, 128] | [128, 128] | [256, 256] | [256, 256] | [128, 128] | [32, 32] |
| hidden activation | tanh | tanh | tanh | relu | relu | relu | relu |
| orthogonal initialization | Yes | No | Yes | fanin | fanin | Yes | Yes |
| initial std | 1.0 | 0.5 | 1.0 | 1.0 | 1.0 | 1.0 | 1.0 |
| | | | | | | | |
| Movement Primitive (MP) type | n.a. | n.a. | n.a. | n.a. | n.a. | ProDMPs | ProDMPs |
| number basis functions | n.a. | n.a. | n.a. | n.a. | n.a. | 8 | 5 |
| weight scale | n.a. | n.a. | n.a. | n.a. | n.a. | 0.1 | 0.1 |
| goal scale | n.a. | n.a. | n.a. | n.a. | n.a. | 0.1 | 0.1 |

[1]Linear Schedule from 0.3 to 0.01 during the first 25% of the training. Then continued with 0.01.

Table 3: Hyperparameters for the Box Pushing Dense, Episode Length $T = 100$

|  | PPO | gSDE | TRPL | SAC | PINK | TCE | BBRL | BBRL Cov. |
|---|---|---|---|---|---|---|---|---|
| number samples | 48000 | 80000 | 48000 | 8 | 8 | 152 | 152 | 152 |
| GAE $\lambda$ | 0.95 | 0.95 | 0.95 | n.a. | n.a. | 0.95 | n.a. | n.a. |
| discount factor | 1.0 | 1.0 | 1.0 | 0.99 | 0.99 | 1.0 | 1.0 | 1.0 |
| | | | | | | | | |
| $\epsilon_\mu$ | n.a. | n.a. | 0.005 | n.a. | n.a. | 0.05 | 0.1 | 0.05 |
| $\epsilon_\Sigma$ | n.a. | n.a. | 0.00005 | n.a. | n.a. | 0.0005 | 0.00025 | 0.0005 |
| trust region loss coef. | n.a. | n.a. | 10 | n.a. | n.a. | 1 | 10 | 10 |
| | | | | | | | | |
| optimizer | adam | adam | adam | adam | adam | adam | adam | adam |
| epochs | 10 | 10 | 20 | 1 | 1 | 50 | 20 | 20 |
| learning rate | 5e-5 | 1e-4 | 5e-5 | 3e-4 | 3e-4 | 3e-4 | 3e-4 | 3e-4 |
| use critic | True | True | True | True | True | True | True | True |
| epochs critic | 10 | 10 | 10 | 1 | 1 | 50 | 10 | 10 |
| learning rate critic | 1e-4 | 1e-4 | 1e-4 | 3e-4 | 3e-4 | 1e-3 | 3e-4 | 3e-4 |
| number minibatches | 40 | n.a. | 40 | n.a. | n.a. | n.a. | n.a. | n.a. |
| batch size | n.a. | 2000 | n.a. | 512 | 512 | n.a. | n.a. | n.a. |
| buffer size | n.a. | n.a. | n.a. | 2e6 | 2e6 | n.a. | n.a. | n.a. |
| learning starts | 0 | 0 | 0 | 1e5 | 1e5 | 0 | 0 | 0 |
| polyak_weight | n.a. | n.a. | n.a. | 5e-3 | 5e-3 | n.a. | n.a. | n.a. |
| SDE sampling frequency | n.a. | 4 | n.a. | n.a. | n.a. | n.a. | n.a. | n.a. |
| entropy coefficient | 0 | 0.01 | 0 | auto | auto | 0 | 0 | 0 |
| | | | | | | | | |
| normalized observations | True | True | True | False | False | True | False | False |
| normalized rewards | True | True | False | False | False | False | False | False |
| observation clip | 10.0 | n.a. | n.a. | n.a. | n.a. | n.a. | n.a. | n.a. |
| reward clip | 10.0 | 10.0 | n.a. | n.a. | n.a. | n.a. | n.a. | n.a. |
| critic clip | 0.2 | 0.2 | n.a. | n.a. | n.a. | n.a. | n.a. | n.a. |
| importance ratio clip | 0.2 | 0.2 | n.a. | n.a. | n.a. | n.a. | n.a. | n.a. |
| | | | | | | | | |
| hidden layers | [512, 512] | [256, 256] | [512, 512] | [256, 256] | [256, 256] | [128, 128] | [128, 128] | [128, 128] |
| hidden layers critic | [512, 512] | [256, 256] | [512, 512] | [256, 256] | [256, 256] | [256, 256] | [256, 256] | [256, 256] |
| hidden activation | tanh | tanh | tanh | tanh | tanh | leaky_relu | leaky_relu | leaky_relu |
| orthogonal initialization | Yes | No | Yes | fanin | fanin | Yes | Yes | Yes |
| initial std | 1.0 | 0.05 | 1.0 | 1.0 | 1.0 | 1.0 | 1.0 | 1.0 |
| | | | | | | | | |
| MP type | n.a. | n.a. | n.a. | n.a. | n.a. | ProDMPs | ProDMPs | ProDMPs |
| number basis functions | n.a. | n.a. | n.a. | n.a. | n.a. | 8 | 8 | 8 |
| weight scale | n.a. | n.a. | n.a. | n.a. | n.a. | 0.3 | 0.3 | 0.3 |
| goal scale | n.a. | n.a. | n.a. | n.a. | n.a. | 0.3 | 0.3 | 0.3 |

Table 4: Hyperparameters for the Box Pushing Sparse, Episode Length $T = 100$

| | PPO | gSDE | TRPL | SAC | PINK | TCE | BBRL | BBRL Cov. |
|---|---|---|---|---|---|---|---|---|
| number samples | 48000 | 80000 | 48000 | 8 | 8 | 76 | 76 | 76 |
| GAE $\lambda$ | 0.95 | 0.95 | 0.95 | n.a. | n.a. | 0.95 | n.a. | n.a. |
| discount factor | 1.0 | 1.0 | 1.0 | 0.99 | 0.99 | 1.0 | 1.0 | 1.0 |
| $\epsilon_\mu$ | n.a. | n.a. | 0.005 | n.a. | n.a. | 0.05 | 0.1 | 0.05 |
| $\epsilon_\Sigma$ | n.a. | n.a. | 0.00005 | n.a. | n.a. | 0.0005 | 0.00025 | 0.001 |
| trust region loss coef. | n.a. | n.a. | 10 | n.a. | n.a. | 1 | 10 | 10 |
| optimizer | adam | adam | adam | adam | adam | adam | adam | adam |
| epochs | 10 | 10 | 20 | 1 | 1 | 50 | 20 | 20 |
| learning rate | 5e-4 | 1e-4 | 5e-5 | 3e-4 | 3e-4 | 3e-4 | 3e-4 | 3e-4 |
| use critic | True | True | True | True | True | True | True | True |
| epochs critic | 10 | 10 | 10 | 1 | 1 | 50 | 10 | 10 |
| learning rate critic | 1e-4 | 1e-4 | 1e-4 | 3e-4 | 3e-4 | 3e-4 | 3e-4 | 3e-4 |
| number minibatches | 40 | n.a. | 40 | n.a. | n.a. | n.a. | n.a. | n.a. |
| batch size | n.a. | 2000 | n.a. | 512 | 512 | n.a. | n.a. | n.a. |
| buffer size | n.a. | n.a. | n.a. | 2e6 | 2e6 | n.a. | n.a. | n.a. |
| learning starts | 0 | 0 | 0 | 1e5 | 1e5 | 0 | 0 | 0 |
| polyak_weight | n.a. | n.a. | n.a. | 5e-3 | 5e-3 | n.a. | n.a. | n.a. |
| SDE sampling frequency | n.a. | 4 | n.a. | n.a. | n.a. | n.a. | n.a. | n.a. |
| entropy coefficient | 0 | 0.01 | 0 | auto | auto | 0 | 0 | 0 |
| normalized observations | True | True | True | False | False | True | False | False |
| normalized rewards | True | True | False | False | False | False | False | False |
| observation clip | 10.0 | n.a. | n.a. | n.a. | n.a. | n.a. | n.a. | n.a. |
| reward clip | 10.0 | 10.0 | n.a. | n.a. | n.a. | n.a. | n.a. | n.a. |
| critic clip | 0.2 | 0.2 | n.a. | n.a. | n.a. | n.a. | n.a. | n.a. |
| importance ratio clip | 0.2 | 0.2 | n.a. | n.a. | n.a. | n.a. | n.a. | n.a. |
| hidden layers | [512, 512] | [256, 256] | [512, 512] | [256, 256] | [256, 256] | [128, 128] | [128, 128] | [128, 128] |
| hidden layers critic | [512, 512] | [256, 256] | [512, 512] | [256, 256] | [256, 256] | [256, 256] | [256, 256] | [256, 256] |
| hidden activation | tanh | tanh | tanh | tanh | tanh | leaky_relu | leaky_relu | leaky_relu |
| orthogonal initialization | Yes | No | Yes | fanin | fanin | Yes | Yes | Yes |
| initial std | 1.0 | 0.05 | 1.0 | 1.0 | 1.0 | 1.0 | 1.0 | 1.0 |
| MP type | n.a. | n.a. | n.a. | n.a. | n.a. | ProDMPs | ProDMPs | ProDMPs |
| number basis functions | n.a. | n.a. | n.a. | n.a. | n.a. | 8 | 8 | 8 |
| weight scale | n.a. | n.a. | n.a. | n.a. | n.a. | 0.3 | 0.3 | 0.3 |
| goal scale | n.a. | n.a. | n.a. | n.a. | n.a. | 0.3 | 0.3 | 0.3 |

Table 5: Hyperparameters for the Hopper Jump, Episode Length $T = 250$

|  | PPO | gSDE | TRPL | SAC | PINK | TCE | BBRL | BBRL Cov. |
|---|---|---|---|---|---|---|---|---|
| number samples | 8000 | 8192 | 8000 | 1000 | 1 | 64 | 64 | 64 |
| GAE $\lambda$ | 0.95 | 0.99 | 0.95 | n.a. | n.a. | 0.95 | n.a. | n.a. |
| discount factor | 1.0 | 0.999 | 1.0 | 0.99 | 0.99 | 1.0 | 1.0 | 1.0 |
| $\epsilon_\mu$ | n.a. | n.a. | 0.05 | n.a. | n.a. | 0.1 | n.a. | 0.005 |
| $\epsilon_\Sigma$ | n.a. | n.a. | 0.0005 | n.a. | n.a. | 0.02 | n.a. | 0.00005 |
| trust region loss coef. | n.a. | n.a. | 10 | n.a. | n.a. | 1 | n.a. | 10 |
| optimizer | adam | adam | adam | adam | adam | adam | adam | adam |
| epochs | 10 | 10 | 20 | 1000 | 1 | 50 | 100 | 100 |
| learning rate | 3e-4 | 9.5e-5 | 3e-4 | 1e-4 | 2e-4 | 1e-4 | 1e-4 | 1e-4 |
| use critic | True | True | True | True | True | True | True | True |
| epochs critic | 10 | 10 | 10 | 1000 | 1 | 50 | 100 | 100 |
| learning rate critic | 3e-4 | 9.5e-5 | 3e-4 | 1e-4 | 2e-4 | 1e-4 | 1e-4 | 1e-4 |
| number minibatches | 40 | n.a. | 40 | n.a. | n.a. | n.a. | n.a. | n.a. |
| batch size | n.a. | 128 | n.a. | 256 | 256 | n.a. | n.a. | n.a. |
| buffer size | n.a. | n.a. | n.a. | 1e6 | 1e6 | n.a. | n.a. | n.a. |
| learning starts | 0 | 0 | 0 | 10000 | 1e5 | 0 | 0 | 0 |
| polyak_weight | n.a. | n.a. | n.a. | 5e-3 | 5e-3 | n.a. | n.a. | n.a. |
| SDE sampling frequency | n.a. | 8 | n.a. | n.a. | n.a. | n.a. | n.a. | n.a. |
| entropy coefficient | 0 | 0.0025 | 0 | auto | auto | 0 | 0 | 0 |
| normalized observations | True | False | True | False | False | True | False | False |
| normalized rewards | True | False | False | False | False | False | False | False |
| observation clip | 10.0 | n.a. | n.a. | n.a. | n.a. | n.a. | n.a. | n.a. |
| reward clip | 10.0 | 10.0 | n.a. | n.a. | n.a. | n.a. | n.a. | n.a. |
| critic clip | 0.2 | lin_0.4 | n.a. | n.a. | n.a. | n.a. | n.a. | n.a. |
| importance ratio clip | 0.2 | lin_0.4 | n.a. | n.a. | n.a. | n.a. | n.a. | n.a. |
| hidden layers | [32, 32] | [256, 256] | [32, 32] | [256, 256] | [32, 32] | [128, 128] | [32, 32] | [32, 32] |
| hidden layers critic | [32, 32] | [256, 256] | [32, 32] | [256, 256] | [32, 32] | [128, 128] | [32, 32] | [32, 32] |
| hidden activation | tanh | tanh | tanh | relu | relu | leaky_relu | tanh | tanh |
| orthogonal initialization | Yes | No | Yes | fanin | fanin | Yes | Yes | Yes |
| initial std | 1.0 | 0.1 | 1.0 | 1.0 | 1.0 | 1.0 | 1.0 | 1.0 |
| MP type | n.a. | n.a. | n.a. | n.a. | n.a. | ProDMPs | ProDMPs | ProDMPs |
| number basis functions | n.a. | n.a. | n.a. | n.a. | n.a. | 3 | 3 | 3 |
| weight scale | n.a. | n.a. | n.a. | n.a. | n.a. | 1 | 1 | 1 |
| goal scale | n.a. | n.a. | n.a. | n.a. | n.a. | 1 | 1 | 1 |

Table 6: Hyperparameters for the Table Tennis, Episode Length $T = 300$

| | PPO | gSDE | TRPL | SAC | PINK | TCE | BBRL | BBRL Cov. |
|---|---|---|---|---|---|---|---|---|
| number samples | 48000 | 24000 | 48000 | 8 | 8 | 76 | 76 | 76 |
| GAE $\lambda$ | 0.95 | 0.95 | 0.95 | n.a. | n.a. | 0.95 | n.a. | n.a. |
| discount factor | 1.0 | 1.0 | 1.0 | 0.99 | 0.99 | 1.0 | 1.0 | 1.0 |
| | | | | | | | | |
| $\epsilon_\mu$ | n.a. | n.a. | 0.005 | n.a. | n.a. | 0.005 | 0.004 | 0.005 |
| $\epsilon_\Sigma$ | n.a. | n.a. | 0.0005 | n.a. | n.a. | 0.00025 | 0.000025 | 0.001 |
| trust region loss coef. | n.a. | n.a. | 10 | n.a. | n.a. | 1 | 25 | 25 |
| | | | | | | | | |
| optimizer | adam | adam | adam | adam | adam | adam | adam | adam |
| epochs | 10 | 10 | 20 | 1 | 1 | 50 | 100 | 100 |
| learning rate | 5e-5 | 1e-4 | 5e-5 | 3e-4 | 3e-4 | 3e-4 | 3e-4 | 3e-4 |
| use critic | True | True | True | True | True | True | True | True |
| epochs critic | 10 | 10 | 10 | 1 | 1 | 50 | 100 | 100 |
| learning rate critic | 1e-4 | 1e-4 | 1e-4 | 3e-4 | 3e-4 | 3e-4 | 3e-4 | 3e-4 |
| number minibatches | 40 | n.a. | 40 | n.a. | n.a. | n.a. | n.a. | n.a. |
| batch size | n.a. | 4000 | n.a. | 512 | 512 | n.a. | n.a. | n.a. |
| buffer size | n.a. | n.a. | n.a. | 4e6 | 4e6 | n.a. | n.a. | n.a. |
| learning starts | 0 | 0 | 0 | 1e5 | 1e5 | 0 | 0 | 0 |
| polyak_weight | n.a. | n.a. | n.a. | 5e-3 | 5e-3 | n.a. | n.a. | n.a. |
| SDE sampling frequency | n.a. | 8 | n.a. | n.a. | n.a. | n.a. | n.a. | n.a. |
| entropy coefficient | 0 | 0 | 0 | auto | auto | 0 | 0 | 0 |
| | | | | | | | | |
| normalized observations | True | True | True | False | False | True | False | False |
| normalized rewards | True | True | False | False | False | False | False | False |
| observation clip | 10.0 | n.a. | n.a. | n.a. | n.a. | n.a. | n.a. | n.a. |
| reward clip | 10.0 | 10.0 | n.a. | n.a. | n.a. | n.a. | n.a. | n.a. |
| critic clip | 0.2 | 0.2 | n.a. | n.a. | n.a. | n.a. | n.a. | n.a. |
| importance ratio clip | 0.2 | 0.2 | n.a. | n.a. | n.a. | n.a. | n.a. | n.a. |
| | | | | | | | | |
| hidden layers | [512, 512] | [256, 256] | [256, 256] | [256, 256] | [256, 256] | [256] | [256] | [256] |
| hidden layers critic | [512, 512] | [256, 256] | [512, 512] | [256, 256] | [256, 256] | [256, 256] | [256] | [256] |
| hidden activation | tanh | tanh | tanh | tanh | tanh | tanh | tanh | tanh |
| orthogonal initialization | Yes | Yes | Yes | fanin | fanin | Yes | Yes | Yes |
| initial std | 1.0 | 0.1 | 1.0 | 1.0 | 1.0 | 1.0 | 1.0 | 1.0 |
| | | | | | | | | |
| MP type | n.a. | n.a. | n.a. | n.a. | n.a. | ProDMPs | ProDMPs | ProDMPs |
| number basis functions | n.a. | n.a. | n.a. | n.a. | n.a. | 3 | 3 | 3 |
| weight scale | n.a. | n.a. | n.a. | n.a. | n.a. | 0.7 | 0.7 | 0.7 |
| goal scale | n.a. | n.a. | n.a. | n.a. | n.a. | 0.1 | 0.1 | 0.1 |

