# OpenReview forum: "Open the Black Box: Step-based Policy Updates for Temporally-Correlated Episodic Reinforcement Learning"
_ICLR.cc/2024/Conference — ICLR 2024 poster_

### Official Review · Reviewer_DXh7 · 2023-10-23

**Soundness:** 2 fair
**Presentation:** 3 good
**Contribution:** 2 fair
**Rating:** 6
**Confidence:** 3

**Summary:**

This paper proposes an approach for episodic reinforcement learning. In such a setting, the policy outputs a high-level trajectory instead of
a single action per state. This idea has been extensively explored (e.g., hierarchical reinforcement learning). The main technical innovations of this work are (1) optimizing over trajectory chunks instead of the whole trajectory and (2) modeling the correlation between states observed during the chunks. The paper is extensively evaluated in multiple benchmarks.

**Strengths:**

The approach is technically sound. While the change from previous work (Otto et al. 202; Li et al. 2023) is relatively limited, the proposed innovations are novel.
The extensive experiments show both the advantages and limitations of the approach.

**Weaknesses:**

The main weaknesses I see in the approach are:
1. Integrating over multiple timesteps breaks the markov assumption from which the learning objective in Eq. 8 is derived. This is because, given the multi-state parametrization, a state s_t depends on more than s_{t-1}. This is true not only for the proposed approach but also more generally for every method that introduces a dependency between states (e.g. smoothing). Previous work even noticed that such smoothing can hurt performance (see Smith et al., A Walk in the Park: Learning to Walk in 20 Minutes With Model-Free Reinforcement Learning).
2. The dependence on a hand-picked segment length K. While having some parts of the algorithm heuristically tuned is generally not a problem, I find this parameter to possibly be challenging to pick (due to its dependence on the task) and time-varying. My intuition is that the problem does not appear in the selected experiments due to their simplicity. I guess it will be much more challenging as soon as the time derivative of the reward drastically changes over time.  A cue in this direction is the failure case on the table tennis task, which shows these characteristics (when the ball is close to the end-effector, the robot needs very high-resolution updates, while a longer horizon is sufficient in the waiting times. This problem is not unusual in most robot tasks: for example, in visual navigation or locomotion, the interdependence between states is rarely constant (e.g., when a sudden obstacle appears, one needs to react fast). I think the paper would benefit from studying this aspect in more detail. Could there be a connection between the derivative of the value function and the segment length K? Can it be learned with the task?

A second limitation is in the experimental setup. Specifically:
a) I find some of the results to be difficult to interpret. In Fig. 6b-c, all methods appear to perform similarly in the dense and sparse setting, which I find surprising. Even more so, almost all approaches (including gSDE, which achieves the best performance) perform better with sparse rewards than dense ones. Why is this case? At least for step-wise methods, dense should do better. Why does this happen?
b) I think that results should be reported for a longer number of interactions. In Fig. 5,6, and 7, the methods do not appear to be fully converged. Showing results up to convergence (as in Fig. 5a) would give a better intuition about the approach's strength in comparison to the baselines.

Other relatively minor limitations:
1. The baselines are not defined (e.g., what paper is BBRL?)
2. The comparison of trajectory-based to state-based is not entirely apple to apple since the trajectory methods require an extra low-level controller, possibly specifically tuned to the robot/task. Their data efficiency could be attributed to this prior. In addition, such a low-level controller is difficult to get in some cases, and a simple PD controller won't do (e.g., locomotion). I would mention this disclaimer to the experimental setup.
3. There is a typo in Eq. 5.

**Questions:**

I would like to see a detailed derivation of the objective in the proposed multi-step setup. In addition, I would appreciate a discussion and possibly experiments on the value of K and a more in-depth analysis to justify the experimental setup.

---

> ### Author Response · Authors · 2023-11-18
> **Reply to reviewer DXh7, #1**
>
> ### **Dear Reviewer DXh7,**
>
> Thank you for your valuable and insightful feedback on our work. We have addressed the major concerns you raised to the best of our knowledge. There are still some open questions that we are actively working on, and we are committed to resolving these in the next few days.
>
> ### **[Update Log, Nov. 18.2023]**
>
> Address the major concerns of all reviewers
>
> Upload additional evaluation and ablation study to Supplementary Material
>
> ### **[Update Log, Nov. 22, 2023]**
> Restructure the paper's storyline, and address the remaining concerns from the reviewers
>
> ### **Novelty and Contribution**
>
> We wish to re-emphasize the novelty and significant contribution of TCE, which builds upon the foundational of previous works. TCE represents a distinct algorithmic advancement in Episodic Reinforcement Learning, going beyond mere modifications of existing models. Our work is pioneering in its incorporation of step-based information into the policy update process of Episodic RL [1,2,3,4]—a field that traditionally relied on black box optimization methods. Importantly, TCE maintains the inherent benefits of broad exploration and trajectory smoothness that are characteristic of episodic RL.
>
> ### **Reference of BBRL**
>
> BBRL stands for Deep Black Box Reinforcement Learning, as described in [4], an episodic RL method that learns trajectories directly in the parameter space. We will update our paper to offer a proper introduction of it.
>
> ### **MDP and Learning Objective**
>
> - Our learning framework maintains the Markovian property, as discussed in section 3.1 of [5]. The Markovian property specifically concerns the state transition probability, $p(s_{t+1}, r_{t} | s_{t}, a_{t})$, indicating that the next state and reward depend solely on the current state and action. It does not impose any assumptions on the policy or the decision process. In our policy, we predict the entire trajectory as a sequence of actions at the start of each episode. This approach to action selection is fundamentally similar to other methods, such as repeating actions over consecutive steps [6] and using temporally correlated noise in action sampling [7].
> - We have thoroughly reviewed the paper by [Smith et al]. However, we note that the low-pass filter technique employed in their study is fundamentally different from the approach in our work and other research focused on directly modeling smooth trajectories. One possible failure reason of such a low-pass filter was discussed in the gSDE work [6], as it tends to cancel consecutive pertubations, and will thus lead to poor expoloration.
> - Our learning objective as presented in Equation (8) can be viewed as an approximation of the episodic RL’s learning objective in Equation (1), rather than being derived from other Policy Gradient Method. This approximation is achieved by replacing the trajectory likelihood and advantage with their respective segment-wise versions.

---

> ### Author Response · Authors · 2023-11-18
> **Reply to reviewer DXh7, #1**
>
> ### **Segment Length (Number) and Value Function Learning**
>
> - The segment length, or the number of segments per trajectory, does indeed affect performance. However, we observed that within an optimal range, such as 10-25 segments per trajectory, this parameter is not overly sensitive compared to other hyperparameters. Thus, we use K=25 in all of our experiments. A detailed study on the influence of the number of segments is provided in the updated supplementary material, Section R.5, which was previously reported in Section C.7.
> - The approach we use in TCE for learning value functions aligns with that of standard RL methods, such as PPO. In our model, the segment length K influences only the policy update and does not affect the learning of the value function. This method of leveraging value functions is a key factor in why TCE achieves faster learning speeds compared to BBRL[4] in most tasks we reported. However, it also leads to a similar performance decline in the table tennis task, mirroring the challenges faced by other step-based RL methods that rely on value function learning.
>
> ### **Experiment Settings**
> - Dense and sparse rewards settings
>     We attribute the observed phenomenon to two main factors. Firstly, as previously discussed in [4], sparse reward settings often provide a clearer task description, aligning the reward structure more directly with the success evaluation criteria. Secondly, for some of the step-based algorithms, we employed a discount factor of 1.0, as opposed to the usual 0.99. This adjustment mitigates the task's sparsity, potentially enhancing the performance of these methods. However, since the task is constrained by a limited number of time steps, the episode return remains within a finite range. Despite these considerations, some step-based algorithms still exhibit poorer performance compared to their counterparts in dense reward settings.
>
> - Longer training iterations
>
>     Previsouly, we found a few baelines collapsed after a certain training iterations. And for off-policy methods, due to their update-to-data (UTD) ratio of 1, as suggested in references [6, 7], their training often take 1-2 days in the reported tasks. However, the main challenge is not the collection of samples, but the network updates and the overhead involved in data exchange between the policy and the environments. Considering that off-policy methods are designed for enhanced sample efficiency, their failure to learn effectively from a data volume comparable to that utilized by on-policy methods highlights their inferior performance in the tasks evaluated in the paper.
>
>
> **References**
>
> [1] Peters, J. and Schaal, S., 2008. Reinforcement learning of motor skills with policy gradients. *Neural networks*, *21*(4), pp.682-697.
>
> [2] Kober, J. and Peters, J., 2008. Policy search for motor primitives in robotics. *Advances in neural information processing systems*, *21*.
>
> [3] Peters, J. and Schaal, S., 2006, October. Policy gradient methods for robotics. In *2006 IEEE/RSJ International Conference on Intelligent Robots and Systems* (pp. 2219-2225). IEEE.
>
> [4] Otto, F., Celik, O., Zhou, H., Ziesche, H., Ngo, V.A. and Neumann, G., 2023, March. Deep black-box reinforcement learning with movement primitives. In *Conference on Robot Learning* (pp. 1244-1265). PMLR.
>
> [5] Sutton, Richard S., and Andrew G. Barto. *Reinforcement learning: An introduction*. MIT press, 2018.
>
> [6] Raffin, Antonin, Jens Kober, and Freek Stulp. "Smooth exploration for robotic reinforcement learning." *Conference on Robot Learning*. PMLR, 2022.
>
> [7] Haarnoja, T., Zhou, A., Abbeel, P. and Levine, S., 2018, July. Soft actor-critic: Off-policy maximum entropy deep reinforcement learning with a stochastic actor. In *International conference on machine learning* (pp. 1861-1870). PMLR.
>
> [8] Haarnoja, T., Zhou, A., Hartikainen, K., Tucker, G., Ha, S., Tan, J., Kumar, V., Zhu, H., Gupta, A., Abbeel, P. and Levine, S., 2018. Soft actor-critic algorithms and applications. *arXiv preprint arXiv:1812.05905*.
>
> ================================================
>
> If you have any further questions, please feel free to ask us.
>
> Thank you.
>
> TCE Authors

---

> > ### Comment · Reviewer_DXh7 · 2023-11-23
> > **Thank you for your reply**
> >
> > Thanks for addressing the issues raised in my review. I am happy to increase my score.

---

> ### Author Response · Authors · 2023-11-22
> **We updated the paper manuscript**
>
> Dear reviewer Dxh7,
>
> Since the rebuttal period is almost over, we are eager to know whether our responses have addressed your concerns properly. Here we humbly ask for your feedback at your convenience such that we may still have the chance to discuss with you and chance to improve our final version before the close of the rebuttal window. Thank you!
>
> All the best,
>
> TCE authors

---

> ### Author Response · Authors · 2023-11-23
> **Thank you for your reply**
>
> Dear reviewer DXh7,
>
> Thank you for your reply! We sincerely appreciate your approval of our work.
>
> Best regards,
> TCE Authors

---

### Official Review · Reviewer_psBV · 2023-11-02

**Soundness:** 3 good
**Presentation:** 2 fair
**Contribution:** 2 fair
**Rating:** 6
**Confidence:** 3

**Summary:**

This is an empirical paper that proposes a segment (sub-trajectory) based reinforcement learning method Temporally-Correlated Episodic RL (TCE).

This method adopts the trajectory-based policy representation using Probabilistic Dynamic Movement Primitives (ProDMP) [Li et. al 2023]. It parameterizes the trajectory as a full Gaussian distribution, whose parameters are predicted by the policy network. This enables temporally coherent exploration by sampling trajectories from the Gaussian distribution.

Subsequently, they divide the trajectory into segments, and run trust-region policy gradient based on the segment-wise advantage function estimate and the projection technique in [Otto et al. 2021], to update the trajectory parameter. To simplify the computation of the likelihood, they utilize the pairwise technique from Li et. al (2023).

The efficacy of this approach is demonstrated through a number of robotics manipulation tasks.

**Strengths:**

- Good empirical performance across a variety of tasks and included negative cases like the table tennis with sparse rewards
- The approach is straight-forward and bridges the gap between the extremes of step-based and trajectory-based reinforcement learning.

**Weaknesses:**

1. The terminology “episodic RL” could potentially lead to confusion, as it bears similarity to the concept of “episodic task” contrasted with “continuing task” [Sutton and Barto, 2018]. A more fitting term, such as “sub-trajectory based RL,” may better capture the essence of the work.
2. The paper would benefit from a more organized exposition of its contributions. It appears that applying policy gradient to segments is the primary novel component. And this work integrates techniques from Li et al. (2023) and Otto et al. (2021). A clearer differentiation from these prior works would enhance the clarity of the paper.
3. While hyperparameters are listed in the appendix, the rationale behind their selection remains opaque. Expanding on this would add depth to the methodology.
4. I feel that the current experimental setup does not sufficiently justify certain design choices. Consider incorporating ablation studies to address the following:
    a. The necessity of the differentiable policy projection, as compared to alternatives like those used in PPO.
    b. The benefits of employing full covariance over diagonal variance in the trajectory representation, especially in light of the increased computational cost.
5. Formatting:
    Certain pages in the main pdf are images, making it impossible to do a text search in the pdf and hindering readability
6. In Section 2.2, the claim "The most effective episodic RL approaches directly explore in parameter space" is presented without supporting references.

**Questions:**

Is there a feasible approach to directly represent each segment, rather than representing the entire trajectory?

---

> ### Author Response · Authors · 2023-11-18
> **Reply to reviewer psBV, #1**
>
> ### **Dear reviewer psBV,**
>
> Thank you for your valuable feedback, which has been instrumental in enhancing our paper. We have addressed the major concern you raised. Additionally, we are actively working on some open questions that emerged from your insights.
>
> ### **[Update Log, Nov. 18.2023]**
>
> Address the major concerns of all reviewers
>
> Upload additional evaluation and ablation study to Supplementary Material
>
> ### **[Ongoing works, Nov. 18.2023]**
>
> Re-organizing the paper structure and re-doing a few experiments.
>
> ### **Terminology**
>
> Episodic Reinforcement Learning (ERL) is a well-established concept, previously applied to learning movement primitives in various studies [1, 2, 3, 4]. This approach shifts the search for solutions from the per-step action space to the trajectory parameter space, thereby promoting broader exploration. In this context, entire trajectories are often treated as single data points, and models are typically trained using black-box optimization methods.
>
> TCE, designed for episodic tasks with a finite horizon, similarly generates a single complete trajectory at the start of each episode, which is then followed until the episode concludes. The key distinction of TCE from earlier ERL approaches lies in its utilization of step-based information from sub-segments of the trajectory, enabling more efficient policy updates.
>
> ### **Contribution Statements**
>
> While building upon the foundations laid by previous approaches like ProDMP and TRPL, we believe that TCE stands out as a significant algorithmic advancement in RL. As a novel framework, TCE distinguishes itself by integrating step-based information into the policy updates of episodic RL. This integration not only boosts update efficiency but also maintains trajectory smoothness and supports correlated action exploration, both temporally and across degrees of freedom (DoF). TCE is the first work to successfully merge the benefits of both RL methodologies, marking a novel advancement in the field.
>
> ### **Hyperparameter Selection**
>
> We executed a large-scale grid search to fine-tune key hyperparameters for each baseline method.
> For other hyperparameters, we relied on the values specified in their respective original papers.
> Below is a list summarizing the parameters we swept through during this process.
>
> - BBRL: Policy net size, critic net size, policy learning rate, critic learning rate, samples per itera-
> tion, trust region dissimilarity bounds, number of parameters per movement DoF.
>
> - TCE: Same types of hyper-parameters listed in BBRL, plus the number of segments per trajectory.
>
> - PPO: Policy network size, critic network size, policy learning rate, critic learning rate, batch size,
> samples per iteration.
>
> - TRPL: Policy network size, critic network size, policy learning rate, critic learning rate, batch
> size, samples per iteration, trust region dissimilarity bounds.
>
> - gSDE: Same types of hyper-parameters listed in PPO, together with the state dependent exploration sampling frequency
>
> - SAC/PINK: Policy network size, critic network size, policy learning rate, critic learning rate, alpha learning rate, batch size, Update-To-Data (UTD) ratio.

---

> ### Author Response · Authors · 2023-11-18
> **Reply to reviewer psBV, #1**
>
> ### **Ablation Studies**
>
> We include additional study in the updated supplementary material. Section R.1 highlights the advantages of using full covariance in enhancing trajectory smoothness. Section R.2 examines the capacity of various methods to model movement correlation, with TCE and BBRL Cov demonstrating the greatest flexibility in trajectory sampling. However, it's crucial to note that the ability to model movement correlation is not the only factor contributing to successful performance. The efficiency of policy updates plays a significant role as well, which is a key reason why TCE outperforms BBRL in the majority of tasks we have reported. Similarly, PINK's performance limitations can be attributed to constraints in learning its Q-function, a challenge it shares with SAC. These factors may account for their relatively poorer performance in some tasks.
>
> ### **Formatting Issue**
>
> We apologize for the inconvenience, this problem is fixed in the updated version of the paper.
>
> ### **Approach to directly represent each segment**
>
> Some studies, such as [5], have employed movement primitives to represent individual segments. However, this approach has been shown to underperform compared to methods that model the entire trajectory, like BBRL, as previously discussed in [4].
>
>
> **References**
>
> [1] Peters, J. and Schaal, S., 2008. Reinforcement learning of motor skills with policy gradients. *Neural networks*, *21*(4), pp.682-697.
>
> [2] Kober, J. and Peters, J., 2008. Policy search for motor primitives in robotics. *Advances in neural information processing systems*, *21*.
>
> [3] Peters, J. and Schaal, S., 2006, October. Policy gradient methods for robotics. In *2006 IEEE/RSJ International Conference on Intelligent Robots and Systems* (pp. 2219-2225). IEEE.
>
> [4] Otto, F., Celik, O., Zhou, H., Ziesche, H., Ngo, V.A. and Neumann, G., 2023, March. Deep black-box reinforcement learning with movement primitives. In *Conference on Robot Learning* (pp. 1244-1265). PMLR.
>
> [5] Bahl, Shikhar, et al. "Neural dynamic policies for end-to-end sensorimotor learning." *Advances in Neural Information Processing Systems* 33 (2020): 5058-5069.
>
> ==============================================
>
> If you have any further questions, please feel free to ask us.
>
> Thank you.
>
> TCE Authors

---

> > ### Comment · Reviewer_psBV · 2023-11-21
> >
> > Thank you to the authors for their detailed response.
> >
> > Many of my initial concerns have been addressed and I appreciate the inclusion of new ablation experiments.
> >
> > However, I remain unconvinced about the benefits of utilizing a full covariance matrix in your approach.
> > I feel that it lacks a baseline of TCE with diagonal covariance: it is not evident that the benefits of full covariance in BBRL Cov would extend to TCE.
> >
> > Additionally, I am interested in understanding the training time overhead when using a full covariance compared with that of a diagonal covariance. Could the authors provide details on this?

---

> ### Author Response · Authors · 2023-11-22
> **We added an additional ablation study (TCE Cov vs. TCE Std) in supplementary, section R.7**
>
> Dear reviewer psBV,
>
> Thank you for your feedback. We are glad to note that our efforts have addressed your concerns.
> To address the remaining issue, we have included an additional ablation study in the updated supplementary material, Section R.7. It conducts a comparison of TCE against its STD variant using a factorized Gaussian policy for the parameter. The results clearly indicate underperformance by the std version, and we attribute this to its limited capacity in modeling movement correlations.
>
> Link to supplementary:
> https://openreview.net/attachment?id=mnipav175N&name=supplementary_material
>
> **Computation cost**
>
> In terms of computational load, the main distinctions between cov and std policies are
>
> - Computational Cost in Convex Optimization for Trust Region Enforcement
>
>   As discussed in [1], cov policies require additional computational effort to satisfy constraints like the KL divergence in the trust region enforcement. This is because cov policies include off-diagonal elements representing the 'rotation' of the Gaussian distribution, necessitating more time in the projection process compared to std policies.
>
> - Matrix Inversion During Likelihood Computation
>
>   For a cov policy, inverting the covariance matrix during likelihood computation has a time complexity of $O(n^3)$, where $n$ is the dimension of the parameter vector $w$. In contrast, for a std policy, this operation has a time complexity of $O(n)$.
>
> These factors translate into a 20-50% difference in training time for methods like BBRL when comparing Cov and Std policies.
> However, in the case of TCE, our focus is on leveraging step-based information for policy updates, where the likelihood computation is w.r.t. the trajectory, as opposed to the parameter $w$.
> Here, a std distribution in the parameter space maps to a blocked diagonal covariance matrix in the trajectory space, as shown in section R.2, figure 1c. Despite its simpler appearance, this format maintains the same time complexity for likelihood computation as a full trajectory covariance matrix.
> Thus, the difference in training TCE Std and TCE Cov is minor, arount 10%.
>
> To efficiently manage the likelihood computation in trajectory space, we employ techniques that estimate trajectory likelihood using segment-based counterparts. As a result of this approach, the computation time for TCE is approximately twice that of BBRL.
>
>
> **References**
>
> [1] Otto, Fabian, et al. "Differentiable Trust Region Layers for Deep Reinforcement Learning." International Conference on Learning Representations. 2021.
>
> ======================================================================
>
> Please let us know if you have further concerns to our work.
>
> Thank you,
>
> TCE Authors.

---

> > ### Comment · Reviewer_psBV · 2023-11-23
> >
> > Thanks for the clarifications. I appreciate the effort in adding the ablations.
> > As my concerns have been addressed, I am happy to increase the score.

---

> > > ### Author Response · Authors · 2023-11-23
> > > **Thank you for enhancing your recommendation**
> > >
> > > Dear reviewer psBV,
> > >
> > > thank you for your deep insights that help us showcase the effectiveness of our work. We are glad to hear our effort addressed your remaining concerns. We are encouraged by your approval of our work.
> > >
> > > Best Regards,
> > >
> > > TCE authors

---

### Official Review · Reviewer_ALmz · 2023-11-03

**Soundness:** 3 good
**Presentation:** 3 good
**Contribution:** 2 fair
**Rating:** 6
**Confidence:** 3

**Summary:**

By using parametric trajectory generators such as Movement Primitives (MP), episodic reinforcement learning techniques tackle robot trajectory generation issues by rephrasing them as contextual optimization. Although these methods are efficient in producing smooth trajectories and detecting certain movement correlations, they do not make use of the temporal structure present in trajectories, leading to less sample efficiency. To overcome these problems, the authors provide the Temporally-Correlated Episodic RL (TCE) technique. By sampling multi-second trajectories in a parameterized space, TCE improves exploration efficiency and ensures high-order trajectory smoothness and movement correlation capture. TCE divides the whole trajectory into smaller parts for policy changes, assessing each part according to its unique benefits.

**Strengths:**

1. Figure 3 efficiently summarizes the entire learning framework.

2. The code base and config files are provided for double-blind review, which enhances the credibility of the study.

**Weaknesses:**

1. I suggest the authors to spend more time re-organizing this paper. The overall description is very messy at present. There is no good connection between the various parts, making it difficult to follow the storyline.

2. Only previous works and their limitations are discussed in the BACKGROUND AND RELATED WORKS section without explaining how they relate to your own work. What's more, there are a lot of claims or conclusions without adding corresponding references, especially in the EPISODIC REINFORCEMENT LEARNING section.

3. The writing of this paper should be improved. The current expression is too redundant, especially in the LIKELIHOOD COMPUTATION OF A SAMPLED TRAJECTORY SEGMENTS section. The author should reasonably remove some content to refine it.

4. The simulation experiment results do not verify the effectiveness of the method. Compared with strong step-based methods, the proposed method does not demonstrate the obvious final performance improvement. Although the variance of the proposed method is the lowest in the CONTACT-RICH MANIPULATION section, the gSDE approach achieves better performance in 2 of 3 tasks. Moreover, the BBRL Cov. algorithm achieves a 20% higher success rate with faster convergence in the HITTING TASK.

**Questions:**

The authors are suggested to experiment with some real scenes. The simulation environments now set by the authors can not provide a big enough challenge to verify the robustness of the algorithm.

---

> ### Author Response · Authors · 2023-11-18
> **Reply to reviewer ALmz, #1**
>
> ### **Dear reviewer ALmz,**
>
> We are grateful for the time and effort you have dedicated to reviewing our submission. In response to your valuable feedback, we are currently re-structuring the storyline of our paper to enhance its readability. An updated version will be provided shortly. In the meantime, we have addressed the other concerns and questions you raised to the best of our knowledge, as detailed below.
>
> ### **[Update Log, Nov. 18.2023]**
>
> Address the major concerns of all reviewers.
>
> Upload additional experiment and ablation study to Supplementary Material.
>
> ### **[Update Log, Nov. 22, 2023]**
> Restructure the paper's storyline, and address the remaining concerns from the reviewers
>
> ### **Writing:**
>
> - We agree with you that the storyline and the connection to other works need to be improved. The revised version will be updated shortly.
>
> ### **Experiments:**
>
> - While we acknowledge that certain baselines, such as gSDE in the box-pushing sparse task and BBRL-cov in table tennis, outperform TCE in specific scenarios, we believe that TCE consistently delivers comparable results across most tasks. Notably, in the MetaWorld tasks, TCE not only achieved the highest asymptotic performance but also maintained the most consistent performance profile. This consistency is characterized by successfully solving a majority of tasks and maintaining a high success rate in those it can solve. Although gSDE shows strong results in box-pushing tasks, its overall performance in the MetaWorld's 50 tasks is moderate, and it fails entirely in the table tennis task. In stark contrast, TCE still manages to achieve a commendable success rate of 65% in the latter.
> - Real Scenes
>
>     We agree that comparing these methods in real-world environments would be very useful. However, as mentioned in our paper, many RL baselines have a problem with smooth movements in their trajectories, which could harm real hardware. Following reviewer dLeV's suggestion, we added a new section on trajectory smoothness in the updated supplementary material (Section R.1). This section shows that most step-based RL methods have very low smoothness, even with well-trained policies. On the other hand, using movement primitives has been successful in real tasks, like imitation learning [1] and Sim2Real transfer in RL [2]. Yet, for real-world applications, we believe it essential to first test and confirm the effectiveness of the policy in a simulation environment before transitioning it to real-world scenarios.
>
>
> **References**
>
> [1] Gomez-Gonzalez, Sebastian, et al. "Adaptation and robust learning of probabilistic movement primitives." IEEE Transactions on Robotics 36.2 (2020): 366-379.
>
> [2] Klink, Pascal, Hany Abdulsamad, Boris Belousov, and Jan Peters. "Self-paced contextual reinforcement learning." In Conference on Robot Learning, pp. 513-529. PMLR, 2020.
>
>
> ================================================
>
> If you have any further questions, please feel free to ask us.
>
> Thank you.
>
> TCE Authors

---

> ### Author Response · Authors · 2023-11-22
> **We updated the paper manuscript**
>
> Dear reviewer ALmz,
>
> we followed your valuable feedbacks and updated the content of the paper, especially making the structure of Section 3 more concise.
> As the rebuttal phase is about to finish, we are wondering if our previous responses have addressed your concerns properly. Your feedback will definitely help us reach a more reasonable decision on our submission.
>
> =======================================================================
>
> The main updates in the manuscript that related to your feedback:
>
> - Revise the abstract and restructure the introduction section to explicitly define the concepts of Step and Episodic RL.
> - Clarify the term "Open the Black Box" in the title, as well as the research gap that we want to solve.
> - Emphasize the main contribution of the paper, and discuss how our framework differs from the existing ones.
> - Split the previous Section 2 "Background and Related Works" into
>   - a) "Preliminaries" as Section 2, focusing on the methods foundational to our framework.
>   - b) "Related Works" as a new Section 4, discussing literature approaches that address similiar research gaps.
> - Link Episodic RL to the MDP in Section preliminaries.
> - Review and refine the claims at the end of Section 2.1, ensuring they are supported by relevant references.
> - For each preliminary method discussed in Section 2, clearly state the specific problem it targets and its relevance to Episodic RL.
> - Provide detailed explanations of the mathematical variables used in our framework, including their dimensions.
> - Remove redundant content in Section 3 and streamline its structure for better clarity and conciseness.
> - Include a discussion on the works closely related to ours, with a special focus on the development of Episodic RL
> - Discuss BBRL (Deep Black Box Reinforcement Learning) in detial as it is highly relevant to our work.
>
> =======================================================================
>
> Additionally, we have included an algorithm box to help understand the paper, along with other newly added evaluations and ablation studies in the updated supplementary material.
>
> Link to updated paper: https://openreview.net/pdf?id=mnipav175N
>
> Link to supplementary material: https://openreview.net/attachment?id=mnipav175N&name=supplementary_material
>
> =======================================================================
>
>
> Thank you!
>
> All the best,
>
> TCE Authors

---

### Official Review · Reviewer_dLeV · 2023-11-06

**Soundness:** 2 fair
**Presentation:** 1 poor
**Contribution:** 1 poor
**Rating:** 6
**Confidence:** 3

**Summary:**

The paper proposes a framework called Temporally-Correlated Eposodic RL, which is able to generate smooth trajectories and capture the movement correlation. In particular, the contributions are:
A. Change the action space to the parameter space of the smooth trajectories.
B. Reconstruct the temporal correlation using pair information.
C. Complete trajectory sampling.
D. Exact likelihood computation for trajectory segments.

**Strengths:**

1. The new parameterization of the trajectory seems to guarantee smoothness by construction.

**Weaknesses:**

1. The novelty of the algorithm is incremental. It is a direct application of Li et al. 2023 to the RL regime, by (1) changing the action space to the parameter space of trajectories, and (2) enlarging the timestep to the segmentation scale.

2. Many terms in the paper are described with words, which makes it hard to understand. I strongly suggest the authors provide mathematical definitions. For example,

- a. Since this is still an RL approach, define the Markov Decision Process first. What is the action space? What is the state space? Now you can explain how your approach differs from traditional RL more clearly, by providing us the details in this different MDP. A tricky question here is that since MDP is Markovian, why do you need the temporal correlation?

- b. It would be more helpful to define the dimensions of $y$, $w$, $\Sigma$ and $s$.

- c. What are d_mean and d_cov in Equation (4)? Are they Frobenius distances?

3. As a reader, I feel many unimportant texts are taking too many places, which makes it not easy to understand the paper. For example, Sections 3.1, 3.2, and 3.4 don't need so many places to mention, since they are merely (a) complete-trajectory sampling, (b) GAE, and (c) TRPO-like. Instead, providing the pseudo-code for the whole learning process would be more valuable as a complement to Figure 3. Currently, there are too many technical things and curves in Figure 3, which makes it hard to understand the learning procedure from this single figure.

4. The result is not sufficient to show the significance of the proposed method. In particular,

- a. After checking Appendix C.2, I found that the baselines perform generally well when TCE works well. However, there exist some situations where TCE significantly fails and the other baselines still work well, and these situations are not rare, including LeverPull, StickPull, Hammer, ButtonPressTopdown, and ButtonPressTopdownWall.

- b. Furthermore, the curves for SAC and PINK are terminated after 10M and 27M interactions respectively, whereas the other methods terminate at 50M interactions, which is an unfair comparison. It is hard to show the efficiency of this algorithm unless these baselines are finished. The reason is: in many cases, these baselines would raise their success rates nonlinearly during the training process, and we cannot conclude by only inspecting their intermediate performances.

- c. Several key ablation studies are missing to test the soundness of each proposed component. See the questions.

Finally, there are some typos in the paper. For example, in Equation 5, the left curly brace is missing. In Equation 6, the (|s) looks strange, it might be (s) instead.

**Questions:**

1. What does black box in the title stand for? Can you help me accurately define it? Does it mean the dynamics is black-box, or the policy approximator (neural network) is black-box? Though this term is repeated several times in the paper, there is no accurate definition in the paper. Furthermore, since the title says 'open the black box', is your algorithm a white-box algorithm?

2. Is Figure 1 a real experiment, or just an illustrative explanation?

3. It seems that Section 3.1 is just a description of whole-trajectory sampling. Is there any novelty in terms of the exploration here? Doesn't normal RL actor also sample using their own stochastic policy?

4.
> Notably, the off-policy methods SAC and PINK were trained with less samples than used for on-policy methods due to their limitations in parallel environment utilization.

      What does the limitation refer to here? Why cannot SAC and PINK be deployed to parallel training? (Also, 'fewer' samples, not 'less' samples).

5. In terms of the ablation studies:

- a. The PPO and SAC in this paper are only used to generate low-level actions. However, both of them are general RL frameworks, which should also be applicable even if the action space is changed to the parameter space of the trajectory. And these versions in the parameter space of trajectory are missing as the baselines. You can still claim that these new baselines are just some naive version of TCE since they do not really calculate per-segment credit assignment and there's no exact probability calculation. But, showing how these frameworks work in this new parameter space is still important, since now we can evaluate how this different action space could improve the sample efficiency by itself.

- b. Smoothness comparison. Define a metric for smoothness, and evaluate how all the final trajectories for all the methods perform under this metric. Since you claim that TCE helps the smoothness, then only showing the success rate is not sufficient.

- c. The significance of the exact probability calculation. If you abandon this component and just use the normal p(w) instead of p(y), how would the performance change empirically?

- d. One thing I am wondering is how this temporal correlation helps the sampling complexity. Can you provide the final Sigma for each DoF? Ideally, this matrix should be far from the diagonal matrix, otherwise, we can just use the step-wise algorithm.

---

> ### Author Response · Authors · 2023-11-18
> **Reply to reviewer dLeV, #1**
>
> ### **Dear Reviewer dLeV,**
>
> Thank you for your insightful review of our paper. Your feedback has been instrumental in enhancing our work. We have addressed the majority of your concerns and added supplementary material with new experiments and ablations. Currently, we are working on addressing the remaining questions and are in the process of restructuring the paper.
>
>
> **[Update Log, Nov. 18.2023]**
>
> Address the major concerns of all reviewers.
>
> Upload additional evaluation and ablation study to Supplementary Material
>
>
> **[Update Log, Nov. 22, 2023]**
>
> Restructure the paper's storyline, and address the remaining concerns from the reviewers
>
>
> ### **Episodic RL & Black-Box Optimization**
>
> Episodic Reinforcement Learning (RL), as discussed in references [1,2,3,4], is a distinct branch in the evolution of RL that emphasizes maximizing the return of an entire episode, rather than focusing on the internal dynamics of the environment. This approach shifts the search for solutions from the per-step action space to the trajectory parameter space, thereby promoting broader exploration. In this context, entire trajectories are often treated as single data points, and models are typically trained using black-box optimization methods [5,6,7,8]
>
> The term 'black box' in our title reflects the essence of these methods, rooted in black-box optimization principles, which tend to overlook the detailed step-based information acquired during interactions with the environment. While these methods excel in expansive exploration and trajectory smoothness, they generally require a larger number of samples for effective policy training. In contrast, step-based RL methods have shown substantial improvements in learning efficiency by leveraging this step-based information.
>
> ### **Novelty & Opening the Black Box**
>
> Despite the leverage of previous approaches, we believe TCE has made a distinct algorithmic advancement in the field of RL. As a novel framework, TCE uniquely incorporates step-based information into the policy updates of episodic RL, simultaneously preserving trajectory smoothness and facilitating correlated action exploration both temporally and across DoF. Thus, by effectively integrating the strengths of the two RL methodologies, TCE successfully 'opens the black box’.
>
> ### **Writing**:
>
> - Figure 1 is an illustration, not an experiment
> - We will add a MDP description in the updated paper. However, our learning framework maintains the Markovian property, as discussed in the Chapter 3.1 of [9]. The Markovian property specifically concerns the state transition probability, $p(s_{t+1}, r_{t} |~s_{t}, a_{t})$, indicating that the next state and reward depend solely on the current state and action. It does not impose any assumptions on the policy or the decision process.
> - d_mean and d_cov are general notations for discrepancy measures, we specifically use KL-divergence in this work as it has been shown to be effective for policy learning within the parameter space of Motion Primitives [5].
>
> ### **Performance against Baselines**
> While we recognize that the performance gains in certain environments may be marginal, it is crucial to highlight that our algorithm, TCE, consistently demonstrates superior or at least comparable results in the majority of tasks when benchmarked against robust baselines. A notable example is the table tennis task, where gSDE failed entirely, yet our algorithm achieved a 60% success rate. We acknowledge TCE's underperformance in 5 out of the 50 Metaworld tasks. However, considering the wide variety of task types in Metaworld, this is not unusual. Similar to other baseline methods, which also exhibit variable performance across different tasks, TCE has not been specifically optimized for any individual task. Despite this, it has shown exceptional overall performance across the comprehensive Metaworld benchmark. This not only underscores TCE's robustness but also its adaptability, making it a highly effective method in a broad spectrum of scenarios.
>
> ### **Training horizon of off-policy methods**
> The original SAC employs an update-to-data (UTD) ratio of 1, as suggested in references [10], a configuration known for balancing training stability with sample efficiency. However, this ratio does not leverage the advantages of parallel processing across multiple environments. The main challenge is not the collection of samples, but the network updates and the overhead involved in data exchange between the policy and the environments. This issue often restricts training to a relatively lower number of samples, typically ≤ 10M steps, as described in [10]. Considering that off-policy methods are designed for enhanced sample efficiency, their failure to learn effectively from a data volume comparable to that utilized by on-policy methods highlights their inferior performance in the tasks evaluated in the paper.

---

> ### Author Response · Authors · 2023-11-18
> **Reply to reviewer dLeV, #1**
>
> ### **Ablation Studies:**
>
> - We include a smoothness evaluation in the updated supplementary material, Section R.1. The evaluation is conducted using three common smoothness metrics in robotics and motion planning. The comparison shows TCE and the BBRL Cov have achieved the highest smoothness among all the approaches.
> - In Section R.2 of the updated supplementary material, we present a qualitative evaluation of the action correlation for all methods. This evaluation highlights the varying capacities of these methods in modeling movement correlations. Both TCE and BBRL Cov successfully model correlations across both time and inter-DoF. In contrast, the original BBRL and PINK are limited to modeling correlations within individual DoF. gSDE demonstrates the capability to model temporal correlations, but only over a few consecutive time steps. Notably, PPO, TRPL, and SAC do not engage in action correlation modeling.
>
>     However, it is important to emphasize that the capability to model movement correlation is not the sole determinant of successful performance. The BBRL, for instance, is hindered by an inefficient policy update strategy. Similarly, PINK's performance limitations can be attributed to constraints in learning its Q-function, a challenge it shares with SAC. These factors may account for their relatively poorer performance in some tasks.
>
> - We agree that including tests for PPO and SAC with Movement Primitives (MP) action space is an important ablation study. Previous work [5] has already shown the ineffectiveness of combining PPO with Movement Primitives (MP).
>
>     In the updated supplementary material, Section R.3, we include an additional ablation study where we combined SAC with MP parameters as the action space. The results from this study suggest that this combination is not effective. We believe this ineffectiveness arises from the increased dimensionality of the action space, which significantly complicates the learning process for the Q-function.
>
> - Using p(w) instead of p(y) for likelihood is exactly what BBRL [5] did. We already included that as a baseline, but we agree that more discussion of the method should be included in the experiment section to clarify the differences, and we are currently working on that.
>
> **References:**
>
> [1] Whitley D, Dominic S, Das R, Anderson CW. Genetic reinforcement learning for neurocontrol problems. Machine Learning. 1993 Nov;13:259-84.
>
> [2] Igel, C., 2003, December. Neuroevolution for reinforcement learning using evolution strategies. In *The 2003 Congress on Evolutionary Computation, 2003. CEC'03.* (Vol. 4, pp. 2588-2595). IEEE.
>
> [3] Peters, J. and Schaal, S., 2008. Reinforcement learning of motor skills with policy gradients. *Neural networks*, *21*(4), pp.682-697.
>
> [4] Kober, J. and Peters, J., 2008. Policy search for motor primitives in robotics. *Advances in neural information processing systems*, *21*.
>
> [5] Otto, F., Celik, O., Zhou, H., Ziesche, H., Ngo, V.A. and Neumann, G., 2023, March. Deep black-box reinforcement learning with movement primitives. In *Conference on Robot Learning* (pp. 1244-1265). PMLR.
>
> [6] A. Abdolmaleki, D. Sim˜oes, N. Lau, L. P. Reis, and G. Neumann. Contextual direct policy
> search. Journal of Intelligent & Robotic Systems, 96(2):141–157, 2019.
>
> [7] Salimans, J. Ho, X. Chen, S. Sidor, and I. Sutskever. Evolution strategies as a scalable
> alternative to reinforcement learning. arXiv preprint arXiv:1703.03864, 2017.
>
> [8] Tangkaratt, V., Van Hoof, H., Parisi, S., Neumann, G., Peters, J. and Sugiyama, M., 2017, February. Policy search with high-dimensional context variables. In *Proceedings of the AAAI Conference on Artificial Intelligence* (Vol. 31, No. 1).
>
> [9] Sutton, Richard S., and Andrew G. Barto. Reinforcement learning: An introduction. MIT press, 2018.
>
> [10] Haarnoja, T., Zhou, A., Abbeel, P. and Levine, S., 2018, July. Soft actor-critic: Off-policy maximum entropy deep reinforcement learning with a stochastic actor. In *International conference on machine learning* (pp. 1861-1870). PMLR.
>
>
> ================================================
>
> If you have any further questions, please feel free to ask us.
>
> Thank you.
>
> TCE Authors

---

> ### Author Response · Authors · 2023-11-22
> **To reviewer dLeV**
>
> Dear reviewer dLeV,
>
> We updated the paper manuscript following your valuable feedback.
>
> As the rebuttal period is near to finish, we are eager to know if our recent responses have sufficiently addressed your concerns. Your insights will definitely guide us towards a more informed decision regarding our submission. We appreciate your feedback and thank you for your time.
>
> All the best,
>
> TCE Authors

---

> > ### Comment · Reviewer_dLeV · 2023-11-22
> > **Thank you for your reply**
> >
> > Thank you for your great effort, and I think most of my concerns are addressed well. Typically, I see
> > - the temporal correlation shows the correlation matrix is not diagonal (if possible, please provide a numerical table showing how far it is from the diagonal matrix when the revision phase begins)
> > - the proposed method outperforms during the smoothness experiment
> > - the SAC in parameter space underperforms
> > - the writing is being improved (still, a pseudo-code would be appreciated in the future revision)
> > - the technical challenge for running SAC is explained
> >
> > I guess the main concern is still from the complete failure under specific environments in Meta-World (4 out of 50). In these experiments, the proposed method not only underperform, but yields nearly 0% success rates. This phenomenon is rare to see for some standard algorithms, (for example SAC), since we would expect the method could more or less yield some trajectories with non-zero scores, if the environment is densely rewarded. In the future, a small exploration of why the method fails under these specific environments would be greatly appreciated. Knowing the limitations better and providing the analysis more thoroughly would also help the community be more glad to try your method in the long term.
> >
> > The paper itself definitely looks unique due to its orthogonal exploration in providing smooth trajectories. I'm happy to update the score to 6.

---

> > > ### Author Response · Authors · 2023-11-22
> > > **Thank you for enhancing your recommendation**
> > >
> > > Dear reviewer dLeV,
> > >
> > > We are glad to hear that our responses and updates to the paper have addressed your major concerns. Your feedback has also provided us with valuable insights on how to evaluate the effectiveness of our approach from various aspects. We truly appreciate your suggestions.
> > > Regarding the pseudo-code, we actually added an algorithm box in the supplementary material, **Section R.8**. You can click this link to get the latest version of it:
> > >
> > > https://openreview.net/attachment?id=mnipav175N&name=supplementary_material
> > >
> > > Regarding the four MetaWorld tasks, we are equally curious about the challenges they present. We observed that the BBRL method, which closely related to our work, encounters similar difficulties in these tasks. Interestingly, there are discussions in the Metaworld GitHub repository indicating potential issues with the reward configurations for button press family. Notably, two of our unsuccessful tasks are part of this category, suggesting that these potential bugs might be a reason for the failures we've encountered.
> > >
> > > - Discussions to these potential bugs in Metaworld
> > >
> > >   https://github.com/Farama-Foundation/Metaworld/issues/389
> > >
> > >   https://github.com/Farama-Foundation/Metaworld/issues/433
> > >
> > >
> > > Thank you and best regards,
> > >
> > > TCE Authors

---

### Comment · Area_Chair_w4zj · 2023-11-20
**Author-Reviewer Discussion Period Ending November 22**

Hi,

Thanks for your help with the review process!

There are only two days remaining for the author-reviewer discussion (November 22nd). Please read through the authors' response to your review and comment on the extent to which it addresses your questions/concerns.

Best,\
AC

---

### Author Response · Authors · 2023-11-20
**Update Log**

### **Update Log [Nov. 18, 2023]**
- Reply to all reviewers regarding their major concerns
- Update the supplementary material with additional evaluation and ablation study.

  - Add an **trajectory smoothness evaluation** in section **R.1**
  - Add an **action correlation evaluation** in section **R.2**
  - Add an ablation study using **SAC to train movement primitives** in section **R.3**
  - Add an ablation study using **PPO-style trust region to train TCE** in section **R.4**
  - Add an ablation study for the **selection of the number of trajectory segments**, as section **R.5** (previously as C.7)
  - List the **key hyperparameters** mainly tuned for the reported experiments, as section **R.6**

### **Update Log [Nov. 22, 2023]**
-  Add an ablation study comparing **TCE Cov with TCE Std** to supplementay material, as section **R.7**
-  Add an **algortihm Box** to explain the TCE training pipeline to supplementay material, **R.8**
-  Update and re-organize the paper structure (main changes are highlighted in blue color).

### **Update to paper [Nov. 22, 2023]**
- Rewrite the abstract and restructure the introduction section to explicitly define the concepts of Step and Episodic RL.
- Explain the meaning of "Open the Black Box" term in the title, as well as its connection to episodic RL methods.
- Emphasize the main contribution of the paper in terms of the development of Episodic RL.
- Split the previous Section 2 "Background and Related Works" into a) "Preliminaries" as Section 2, and b) "Related Works" as Section 4.
- Add definitions of Markov Decision Processes (MDP), and an explanation of how Episodic RL does not violate the Markov property.
- Add explanations to the mathematical variables used in the framework, as well as their dimensionalities.
- Remove redundant content in Section 3 and make the structure more concise.
- Add a discussion on the works closely related to ours, especially the development of episodic RL, and the method BBRL (Deep Black Box Reinforcement Learning).


Link to updated paper: https://openreview.net/pdf?id=mnipav175N

Link to supplementary material: https://openreview.net/attachment?id=mnipav175N&name=supplementary_material

---

### Meta-Review · Area_Chair_w4zj · 2023-12-12

**Metareview:**

The paper proposes a framework (Temporally-Correlated Episodic RL (TCE)) for reinforcement learning (RL)-based robot trajectory estimation. While existing approaches produce smooth trajectories and maintain movement correlations, they do not exploit the temporal structure within trajectories, which negatively affects their sample efficiency. As a means of addressing this, the paper proposes temporally-correlated episodic RL (TCE) that employs a trajectory-based policy representation based on probabilistic dynamic movement primitives (ProDMP) [Li et al. 2023]. TCE parameterizes the trajectory as a Gaussian distribution the parameters of which are predicted by the policy network. TCE improves the efficiency of exploration while ensuring high-order trajectory smoothness and the preservation of movement correlations. As part of the policy updates, TCE decomposes the full trajectory into smaller components for policy changes, assessing each part according to its unique benefits. The paper evaluates the method on a series of robot manipulation tasks.

The paper was reviewed by four reviewers who as a result of discussions with the author are in consensus regarding their overall assessment of the paper. Initially, the reviewers raised concerns about the significance of the paper's contributions over the work on which it builds (notably that of Li et al. 2023); the insufficiency of the experimental evaluation (Reviewers DXh7, dLeV, and ALmz); and problems with the organization and presentation that made the paper difficult to read. In addition to providing responses to the reviewers, the authors made significant changes to the paper, which the reviewers find to have addressed most of their initial concerns as evidenced by the fact that all four reviewers increased their overall ratings. As one exception, Reviewer dLeV feels that the issue of the complete failure of the method for specific environments (with 0% success rates), was not adequately addressed.

**Justification For Why Not Higher Score:**

Without someone championing the paper, the AC feels that a higher score is not warranted.

**Justification For Why Not Lower Score:**

The paper improved significantly as part of the review process, with all four reviewers increasing their overall ratings, two from a 3 to a 6 and the other two from a 5 to a 6. The reviewers comment that most of their concerns have been addressed.

---

### Decision · Program_Chairs · 2024-01-16

Accept (poster)